

# Geology datasets of North America for use with ice sheet models

Evan J. Gowan[1], Lu Niu[1], Gregor Knorr[1], and Gerrit Lohmann[1]

[1]Alfred Wegener Institute, Helmholtz Centre for Polar and Marine Research, Bremerhaven, Germany

**Correspondence:** Evan J. Gowan (evan.gowan@awi.de)

**Abstract.**

The ice-substrate interface is an important boundary condition for ice sheet modelling. The substrate affects the ice sheet by allowing sliding through sediment deformation and accommodating the storage and drainage of subglacial water. We present three datasets with different geological parameters for the region that was covered by the ice sheets in North America, including Greenland and Iceland. The first dataset includes the distribution surficial sediments, which is separated into continuous, discontinuous and predominantly rock categories. The second dataset includes sediment grain size properties, which is divided into three classes: clay, silt and sand, based on the dominant grain size of the glacial sediments. The third dataset is the generalized bedrock geology. We demonstrate the utility of these datasets for governing ice sheet dynamics by using an ice sheet model with a simulation that extends through the last glacial cycle. Changes in ice thickness by up to 40% relative to a reference simulation happened when the shear friction angle was reduced to account for a weaker substrate. These datasets provide a basis to improve the basal boundary conditions in ice sheet models.

Gowan, E.J., Niu, L., Knorr, G., and Lohmann, G., 2018. Geology datasets of North America for use with ice sheet models, link to datafiles. *PANGAEA*, https://doi.pangaea.de/10.1594/PANGAEA.895889

## 1 Introduction

Temperate ice sheets, such as the Laurentide and Eurasian ice sheets behaved differently depending on whether or not the substrate was covered by thick, continuous unconsolidated sediments (Clark and Walder, 1994). These sediments provided a potential pathway for subglacial water storage and drainage. Areas where crystalline bedrock is predominant at the surface tend to have eskers, indicating that subglacial water drained via large tunnel systems (Clark and Walder, 1994). The subglacial drainage where the surface is covered by continuous, unconsolidated sediments tends to be via linked channel systems (Carlson et al., 2007). The main cause of these different drainage regimes is likely related to the roughness of the bed (*i.e.* in areas with sediment cover, the surface is smoothed by the glacier, while in areas with bedrock outcrops will be more irregular). Sediment deformation in areas with continuous cover is also hypothesized to play a prominent role in the motion of glaciers (Boulton et al., 2001; Evans et al., 2006), possibly also including decoupling with the underlaying, non-deforming surfaces (Kjær et al.,





2006). When sediments become water saturated, they become mechanically weaker than the overlying ice. If this happens, it causes a decoupling from the underlying bed and allows the ice to flow faster than with ice deformation alone. Whether or not this mechanism could have been spatially and temporally pervasive is still open to debate (Piotrowski et al., 2004; Iverson and Zoet, 2015).

5    In North America, there was a distinct difference in ice sheet behavior between the sparsely covered Canadian Shield and the sediment covered sedimentary basins at the southern and western fringes, and Hudson Bay and the Foxe Basins in the center and north. The most striking imprint of this in the geomorphological record is the reduced number of ice streams on much of the Canadian Shield, while areas covered with continuous sediments have many (Margold et al., 2015). The presence or absence influenced the distribution of ribbed moraine, drumlins and eskers on the Canadian Shield (Aylsworth and Shilts, 1989). Retreat of the Laurentide Ice Sheet after the Last Glacial Maximum (LGM) also slowed when the ice sheet became confined to the Canadian Shield (Dyke, 2004). During the advance of the ice sheet prior to the LGM, the margin remained close to the Canadian Shield boundary until the ice sheet reached a threshold that allowed to advance onto the surrounding plains (Dyke et al., 2002). The part of the ice sheet that covered the plains had a low profile relative to the Canadian Shield, which has been attributed to this contrast in basal conditions (Fisher et al., 1985; Licciardi et al., 1998; Gowan et al., 2016). 15    Observations of the dynamics in modern glaciers, for instance in Svalbard, are strongly governed by whether or not they lay on sedimentary or crystalline bedrock (*e.g.* Jiskoot et al., 2000). Areas underlain by sedimentary bedrock are more likely to a thick, fine grained till that is easier to deform when subjected to water saturated conditions.

Having realistic basal conditions is essential in numerical ice sheet modelling. Many ice sheet modelling studies of the Laurentide Ice Sheet (Calov et al., 2002; Tarasov and Peltier, 2004; Gregoire et al., 2012; Abe-Ouchi et al., 2013) used the global 20    sediment thickness map (Laske and Masters, 1997), which was designed for seismology applications rather than surficial processes. This dataset reflects the thickness of Phanerozoic sedimentary rocks that have not undergone significant metamorphism. This map does not reflect the actual distribution of unconsolidated sediments, as many regions of the Canadian Shield do have continuous sediment cover (Aylsworth and Shilts, 1989), and there are regions of discontinuous unconsolidated sediment cover where there is sedimentary bedrock (Fulton, 1995; Soller and Garrity, 2018). This dataset also misses Precambrian sedimentary 25    basins that are overlain by unconsolidated sediments that were modified by ice sheets (Cofaigh et al., 2013). The direct impacts on ice sheet dynamics may only depend on the uppermost few meters of unconsolidated sediment (Boulton et al., 2001; Iverson and Zoet, 2015), so this map may not be representative of the sediment properties that affected the ice sheet. More recently, Stokes et al. (2012) and Tarasov et al. (2012) used a more complete parameterization with additional data from the surficial materials map by Fulton (1995). They use a parameter from 0 (no sediments) to 1 (pervasive sediments). Previous modelling 30    studies did not directly account for variability in the grain size or other properties of the sediments.

In order to gain flexibility in parameterizing sediment parameters for ice sheet modelling, we present three datasets. These data come from existing surficial geological maps when possible, and inferred from other studies where coverage is not complete. (i) The *Sediment distribution dataset* contains information on the distribution of sediment cover, whether continuous, veneer, or dominantly bedrock. (ii) The *sediment grain size dataset* contains information on the average grain size of the sediments. This is based on common geological descriptions of sandy, silty and clay rich diamiction and glacial sediments. (iii)



The *bedrock geology dataset* contains the generalized bedrock type, including distinctions between sedimentary, igneous and metamorphic rocks. These data can be used in a variety of ways, such as by changing the mechanical strength and frictional resistance of the sediment (such as the shear friction angle), effects of hydrology (porosity and permeability of the sediments or rock, type of drainage), roughness of the bed, and the erodibility of substrate.

In order to be usable in ice sheet models, it is necessary to have a continuous dataset. Since existing geological map datasets are discontinuous, due to the presence of post-glacial sediments and water bodies, we had to fill in these gaps. These datasets include supplementary information from geophysical surveys and coring studies to compliment existing maps. We also made an inference on grain size properties in the vast regions without information by using geological maps.

## 2   Description of datasets

### 2.1   Overview and construction

With this dataset, the goal is to represent the subglacial sediment properties for the most recent glaciation, the late Wisconsin glaciation in North America, for use in paleo-ice sheet modelling and reconstruction. The late Wisconsin extended between about 31 000 and 34 000 yr BP (years before present) to about 7000 yr BP (Dyke et al., 2002). For ice sheet modelling, using the modern day distribution and composition of glacial sediments is likely sufficient to use as a boundary condition for the most recent glacial period, though further back in time, this assumption may not be valid (Clark and Pollard, 1998). There are great uncertainties in many of the boundary conditions used in ice sheet modelling, such as uncertainties in past atmospheric and ocean conditions, but sediment cover likely does not change that greatly in a single glaciation (Piotrowski et al., 2001), so we do not feel this is a major setback for the use of this dataset. We want to emphasize that the categories chosen for this dataset are simplified from some of the original data sources in order to make it easier for ice sheet modellers to manipulate a limited range of parameters, rather than match specific geological observations that may only be applicable very small regions. The lack of sediment grain size information over much of Canada also precludes a large range of geological parameters. When ice sheet modelling, it is necessary to have continuous boundary conditions over the whole domain. In areas without geological information, it is necessary to make inferences on the properties based on alternative sources of information, such as bedrock geology maps.

The three datasets are largely based on existing surficial and bedrock geology maps (Table 1). Wherever possible, we used the most up-to-date regional scale (*i.e.* >1:500,000 scale) maps, in order to make it possible to construct the entire dataset in a reasonable amount of time. For the sediment distribution data, where there was overlapping with the map by Fulton (1995), we favoured the more recent dataset. The first step was to import the existing shapefiles of the maps (or digitizing paper maps if not available), and break up the units into the classification schemes that we are using. This involved removing any water bodies and post-glacial sediment units from the maps, and simplifying glacial that had a more complex scheme than we use. The resulting datasets have gaps. To fill in the gaps, we expanded the polygons in a way to favour the dominant unit in the region, or to extend the trend of elongated units.The datasets were edited using ArcGIS and QGIS.



There are many areas where late Wisconsin till is buried by glacio-fluvial and Holocene non-glacial sediments, so the nature or existence of glacial sediments is uncertain. This is also true for previously glaciated areas under lakes and the oceans and places currently covered in glaciers, ice caps and the Greenland Ice Sheet. In these regions, we tried to find published sediment cores, sedimentary sections, and geophysical data that can be used to estimate the the properties of the sediments (Table 2).

We incorporate sediment data from areas outside of the late Wisconsin limit, as in an ice sheet simulation, the exact margin of the ice sheet is unlikely to match the geologically constrained limit, and could become more expansive. For areas south of the Laurentide Ice Sheet limit, there is glacial sediment from more extensive, older glaciations. These data were taken from the US quadrangle maps (Table 1). In other areas such as Alaska and offshore regions, we take the properties from non-glacial sediments and inferences from bedrock geology maps.

In the creation of the dataset, existing shapefile compilations were used if available, which have variable resolution. To simplify the datasets when the originals were high resolution, we used bend simplify tool in the ARCGIS Cartography/generalization Toolbox, with a tolerance of 5 km, and minimum area of 25 km$^2$ (25000000 m$^2$). This is visually similar to the generalization that was used in the surficial materials map by Fulton (1995). Any polygon that had a total area that was less than 2.25 km$^2$ (2250000 m$^2$) was merged to the polygon that had the largest shared border to further simplify the dataset.

## 2.2 Sediment distribution dataset

The map of glacial sediment distribution is shown on Figure 1. Data sources for this dataset are shown in Table 1. By "glacial sediment" we are referring to sediment that is produced as a direct result of glacial action. In a generic sense, it is synonymous with diamiction or till, an unsorted sediment with grain size ranging from clay to boulder. When possible, we try to determine the distribution of glacial sediments in extensive areas covered by post-glacial cover and water bodies (see Table 2). A detailed

explanation for the distribution units, which is based on the scheme found on the Surficial Materials of Canada map by Fulton (1995) is as follows:

**Rock**: Bedrock outcrops are predominant (>75% of the surface area is exposed bedrock (Fulton, 1995)) and extensive glacial sediment deposits are rare. We include "regolith" areas in the northern Canadian archipelago, which were not pervasively affected by late Wisconsin glaciation (even if the upper layer was not well consolidated) and therefore do not produce glacial

sediment deposits.

**Veneer**: Many maps seem to have a different definition on what "veneer" means. In general, it means that glacial sediment deposit are discontinuous (can be zero thickness), but the area covered in glacial sediment exceeds that of exposed rock. The topography of the underlying bedrock is usually visible in these areas. In most maps, these areas have "thin" cover, with thin being defined as anything between less than one meter to as much as ten meters. Commonly, the cutoff is set to be 2-3

meters, although some maps (*e.g.* the Surficial Materials map of Canada by Fulton (1995)) do not explicitly state a value. A recommended thickness value setting for veneer areas should be less than three meters to conform to the most common description of "veneer" provided in maps used in this dataset, though even a thin layer of glacial sediment might affect the dynamics of an ice sheet (Evans et al., 2006).



**Blanket**: These regions are defined as regionally continuous glacial sediment. As with the "veneer" classification, it is not always clear what thickness or distribution is used as a threshold for defining "blanket". If values are given, the threshold is usually greater than three meters. In areas with a blanket of sediment, generally the underlying bedrock topography is not obvious. Glacial sediment units that are described as "hummocky" are included in this definition. These glacial sediments

formed during stagnation of the ice sheet, and are commonly found on elevated regions in western Canada (Eyles et al., 1999). The thickness can vary from a few meters to more than 25 meters, but it is assumed here that these deposits are at least three meters and can be put into the blanket definition.

## 2.3 Sediment grain size dataset

The map of generalized grain size of glacial sediments is shown on Figure 3. To simplify the classification, we only have

10 three main classification types, based on the dominant grain size of the sediments. This classification scheme is based on the Surficial Materials in the Conterminous United States map (Soller and Reheis, 2004), and we attempted to unify this scheme with maps and data in Canada. Glacial sediments tend to be very poorly sorted, so these values should be assessed as being an average composition. A glacial sediment, diamiction or till (the later has a definitive glacial origin) is an unsorted material with grain size ranging from clay to boulder, but the average grain size can vary depending on the local source material. As

an example, in the map by Soller and Reheis (2004), clay rich glacial sediment exists in areas around the Great Lakes, where source material was derived from lake sediments, and sandy glacial sediment exists in mountainous regions where there are extensive rock outcrops.

**clay**: Glacial sediment has a large clay component (>50%).

**silt**: intermediate of clay and sand dominant composition. This unit includes any description called "loamy till", which is a

20 soil with an average grain size between sand and silt.

**sand**: Sand rich till, with the possible presence of abundant coarser grained (pebble to boulder sized) material, with only minor clay component and more sand than silt.

Many maps do not give specific classifications of the grain size of glacial sediments. The United States quadrangle maps (Table 1), which cover most areas south of about 54° north (except in the Cordillera), fortunately do have this information. The

25 lack of information north of this is likely due to accessibility issues, where there are few extensive geology/soil/engineering surveys that would serve as the basis for such a map. As a result, the sediment type for many of these regions was derived from bedrock geology maps. In general, glacial sediments in North America have a composition that similar to the underlying bedrock (Fulton, 1989), so we assume that the grain size should be related to the bedrock geology. Our approach for classifying grain size from geology maps is as follows:

**silt**: fine grained clastic sedimentary rocks (shale, carbonates); mafic igneous rocks; undivided igneous rocks; low grade metamorphic rocks (*e.g.* greenschist)

**sand**: course grained clastic sedimentary rocks (sandstone, conglomerate); felsic igneous rocks; high grade metamorphic rocks (*e.g.* gneiss)



## 2.4 Bedrock geology dataset

This dataset is a simplification of the Geologic Map of North America (Reed et al., 2004; Garrity and Soller, 2009). For the area covered by the Greenland Ice Sheet, we use the map by Dawes (2009). The rocks were divided into the following groups:

**Sedimentary**: All units described as being sedimentary.

**Felsic plutonic**: All rocks explicitly described as felsic igneous (*e.g.* granite), charnockite, units described as being "felsic and intermediate" and units that were undivided mafic and felsic rocks.

**Felsic volcanic**: Same as felsic plutonic, but explicitly described as volcanic (*e.g.* rhyolite)

**Mafic plutonic**: All rocks explicitly described as mafic igneous (*e.g.* gabbro), units described as being "intermediate" and

"intermediate and intermediate", alkaline, and units that were undivided mafic and felsic rocks.

**Felsic volcanic**: Same as plutonic, but explicitly described as volcanic (*e.g.* basalt), also includes volcanic deposits that are described as having interlayered sedimentary layers

**low grade metamorphic**: Marble, plus units described as being "undivided crystalline rocks"

**high grade metamorphic**: Units that are highly metamorphosed *i.e.* gneiss

The map has few units that can be confidently placed in the low grade metamorphic class, because most of these units are grouped with their non-metamorphosed source rock class. Therefore it should be assumed that many of the areas with igneous and sedimentary rocks have undergone some level of metamorphism, particularly on the Canadian Shield. We placed the "undivided unit" in the low grade category, as these most of these areas are in the continental shelf where no geophysical surveys or sampling has taken place. The description given in the original dataset indicates that these rocks likely contain some

amount of metamorphism. In can be assumed that these rocks along the Atlantic coast were probably subjected to some amount of metamorphism during the opening of the Atlantic Ocean, or in the case of Hudson Bay are likely part of the Precambrian Shield.

## 2.5 Caveats

In this compilation, we tried to incorporate the most recent information on surficial geology that was available. Unfortunately,

there are places where, due to discrepancies between adjacent maps, there are visible seams. This is especially evident at the Yukon-Alaska border and the British Columbia - Washington border. Obviously, these areas will be in need of revision when new mapping information becomes available. There are also discrepancies in interpretation and classification between maps. A good example is the dataset we used for Manitoba (Matile and Keller, 2006), which had only two classes for distribution (blanket and rock). The corresponding map by Fulton (1995) divides the regions that are classified as "rock" into veneer and

rock. Since our intention is to use the most up-to-date information, we use the dataset by Matile and Keller (2006), but with the caveat that this also causes a seam with the adjacent regions in northern Ontario and Saskatchewan that has a broader classification scheme.



## 3   Usage in ice sheet models

### 3.1   Parameters

Some general properties of sediment grain size types are shown on Table 3. Most of these properties are described in more detail in Cuffey and Paterson (2010). These properties are only given in a qualitative manner because there have been relatively few in-situ or laboratory measurements of these properties over a range of composition (Iverson and Zoet, 2015). Measured permeability values were reported to be between $10^{13}$–$10^{16}$ m$^2$ (Cuffey and Paterson, 2010). It is recommended that when modelling the behavior of ice sheets, that a range of values be explored.

The effect of sediment distribution on ice sheet models is less well known. The patchiness of sediments may result in "sticky spots", primarily though bedrock nobs that resist the flow of ice (Alley, 1993). The lack of sediment in an otherwise sediment covered region may increase resistance to flow as well (Stokes et al., 2007). The influence later process is likely controlled by the availability of subglacial water. All of the thickness categories made in this dataset are derived from existing geological maps. Because of inconsistencies in classification between maps, and vast regions where there are few direct observations, it is not possible to give a detailed quantitative estimates of distribution or thickness. These exact values of the percentage of surface cover, and sediment thickness can be set as a variable in ice sheet models.

The geological map can be used for determining the erosive properties of the rocks, the source material of glacial sediment (as we did for the grain size dataset), and drainage of water under the ice. For the later case, the transition from Precambrian rock and sedimentary rock has been used to explain the relative absence of eskers south of the Canadian Shield by accommodating the basal meltwater (Grasby and Chen, 2005). Modelling of the effect of bedrock on subglacial water routing has been done by Carlson et al. (2007).

### 3.2   Example of usage of datasets in an ice sheet model

To show the utility of the dataset, we incorporate the information for use with the ice sheet model PISM 1.0 (Bueler and Brown, 2009; PISM authors, 2017), with the addition of an index forcing scheme described in (Niu et al., 2017). In PISM, the basal sediments influence ice sheet dynamics by assuming they deform as a Mohr-Coulomb plastic material (Tulaczyk et al., 2000). The relationship that governs the relationship between the material and the yield stress, $\tau_c$, is:

$$\tau_c = c_o + N \tan(\phi) \tag{1}$$

The sediment parameters include the apparent cohesion, $c_o$, and the shear friction angle, $\phi$. The cohesion is generally regarded as insignificant (Cuffey and Paterson, 2010) and set to zero in most ice sheet simulations (Bueler and Brown, 2009). The shear friction angle is the angle that a material will fracture given a normal stress above its yield strength. This is the primary factor used to tune the basal sediment strength in PISM. In situ and laboratory experimental values of $\phi$ for glacial sediments have a large range, between $18°$ and $40°$ (Cuffey and Paterson, 2010). The parameter $N$ is the difference between the normal stress from the load of the ice sheet and the water pressure in the sediments. In PISM, this factor is generally high



enough that the sediments will not deform unless they are saturated. In our simulations, $N = 0.01$ when saturated. For the tests of these datasets, we only adjust $\phi$.

The results shown below are for an ice sheet model that is run for the entire last glacial cycle, the past 122 000 years. A time slice at 21 000 yr BP is chosen to display the effect of changing the sediment friction angle, as this was when the North American ice sheets were near their maximum extent (Dyke, 2004). Niu et al. (2017) provide a full description of the parameters related to other boundary conditions. The shear friction angle used in their study was a constant $30°$, so to show the effects of changes in basal geological parameters, this value is lowered. The results given below are just to show the effects of changing the basal properties, we make no recommendation of what the values should be. Ultimately, the model used in PISM is dependent on having enough water produced to saturate the sediment layer (Bueler and Brown, 2009). If the water production is too low (*i.e.* the basal temperature of the ice is below pressure melting point), changing the shear friction angle will have no effect on the simulation. Therefore, in the cases shown in this section, the largest changes occur in places where there is significant ice flow, or are connected to ocean basins. Efforts to combine the effects of these datasets with ice sheet hydrology and ice dynamics are ongoing, and show that this model substantially underestimates that amount of water that should be available at the base (Gowan et al., 2018).

The basal conditions model in PISM is based on the assumption that the entire base of the ice sheet is covered in potentially deformable sediments, the strength of which is controlled by the sediment shear friction angle. A lower angle will weaken the ice-bed interface, and therefore encourage sliding. The philosophy of the choice of shear friction angle in these examples is as follows. Areas with continuous sediment cover should be weak, since sediment deformation will be the dominant factor in sliding. The angle in sediment covered areas are lowered from the reference value to accommodate this. For the grain size data, finer grained sediments will be weaker than coarse grained sediments, so the angle in areas with finer sediments are lowered from the reference value. For the geology dataset, we expect that areas underlain with sedimentary and mafic volcanic rocks will be more prone to erosional effects, and therefore more likely to produce unconsolidated sediments, and should therefore be weaker. The angle in these areas are reduced from the reference to simulate this effect.

### 3.2.1 Impact of sediment distribution

The basal boundary condition in PISM has an assumption that continuous sediment cover is over the entire domain (Bueler and Brown, 2009). In order to simulate the differences in sediment distribution, the shear friction angle is changed depending on the coverage. For continuous areas, it is set to $\phi = 10°$ (weak, deformable bed), for discontinuous areas it is set to $\phi = 20°$, and for dominantly rock it is set to $\phi = 30°$ (strong, undeformable bed). Using these values, most of the Canadian Shield has a shear friction angle of $20°$, while areas underlain by Phanerozoic sedimentary rocks have values of $10°$. The impact of this is that there are reductions of ice along the east coast of Canada, the Cordillera, the Great Lakes region, and western Arctic Archipelago and Greenland by up to 40% (Figure 4). There is also an increase in ice thickness in the area east of the Cordillera (5-10% greater), south of the Great Lakes and in Hudson Strait. The lower resistance to flow likely leads the ice sheet to flow further south of the Great Lakes relative to the default simulation. The lack of change in the Canadian Shield, despite decreasing the shear friction angle, is most likely due to the lack of meltwater production to cause a reduction in basal strength.



### 3.2.2 Impact of sediment grain size

To test the effects of sediment grain size type, the input map from Figure 2 was converted to a shear friction angle input by setting clay to $\phi = 10°$, silt to $\phi = 20°$ and sand to $\phi = 30°$. This simulates the fact that clay rich sediments are mechanically weaker, even though an angle of $\phi = 10°$ is below the low end of measurements of real till (Cuffey and Paterson, 2010). The difference in ice thickness at 21 000 yr BP is shown on Figure 5. In this case, most of the Canadian Shield, Greenland and parts of Cordillera have a shear friction angle of $30°$, some areas south of the Great Lakes are $10°$, while the rest is $20°$. The end result at 21 000 yr BP is that there is less change in the simulation compared to the reference. There is a slight reduction in ice thickness in the Cordillera (10-20%) and east coast of Canada (5-10%). South of the Great Lakes, where there is clay rich till with an angle of $10°$, the ice sheet goes further south than the reference simulation.

### 3.2.3 Impact of bedrock geology

The effects of bedrock geology are shown on Figure 6. For this simulation, we adjusted the shear friction angle downwards for geological types that are more likely erode to produce deformable sediments. Sedimentary rocks are given an angle of $\phi = 10°$, to indicate their relative weakness. Low grade metamorphic rocks (which includes areas where the geology is uncertain), are given an angle of $\phi = 20°$. Volcanic rocks are assigned a value of $\phi = 25°$, as they should be more likely to be erodible than plutonic rocks. Plutonic rocks and high grade metamorphic rocks retain the default value of $\phi = 30°$. The results show a decrease in ice thickness in the Cordillera, Canadian Archipelago, eastern Canada and northeastern Greenland by up to about 30%. These areas are largely underlain by sedimentary rocks. As with the other simulations, south of the Great Lakes region, the ice sheet goes further south than the reference simulation.

## 4 Conclusions

We have presented three datasets that present different types of geological data, including sediment distribution, grain size, and bedrock geology. These datasets are intended for use in ice sheet models, where the geological parameters will have impacts on ice sheet dynamics and hydrology. We demonstrated that changing the basal conditions in an ice sheet model on the basis of these datasets do impact the thickness of the ice. With these datasets, we hope that improvements can be made to ice sheet models to incorporate this geological data and gain a better understanding of basal conditions.

## 5 Data availability

Gowan, E.J., Niu, L., Knorr, G., and Lohmann, G., 2018. Geology datasets of North America for use with ice sheet models, link to datafiles. *PANGAEA*, https://doi.pangaea.de/10.1594/PANGAEA.895889



*Author contributions.* EJG compiled the datasets and was the main author of the text. LN designed the ice sheet model simulation. All authors contributed to the text and design of the study.

*Competing interests.* There are no competing interests.

*Acknowledgements.* This work was funded by the Helmholtz Climate Initiative REKLIM (Regional Climate Change), a joint research project
of the Helmholtz Association of German research centres (HGF). This study was also supported by the PACES-II program at the Alfred
Wegener Institute and the Bundesministerium für Bildung und Forschung funded project, PalMod. Development of PISM is supported by
NASA grant NNX17AG65G and NSF grants PLR-1603799 and PLR-1644277. Figures in this paper were plotted with the aid of Generic
Mapping Tools (Wessel et al., 2013).



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



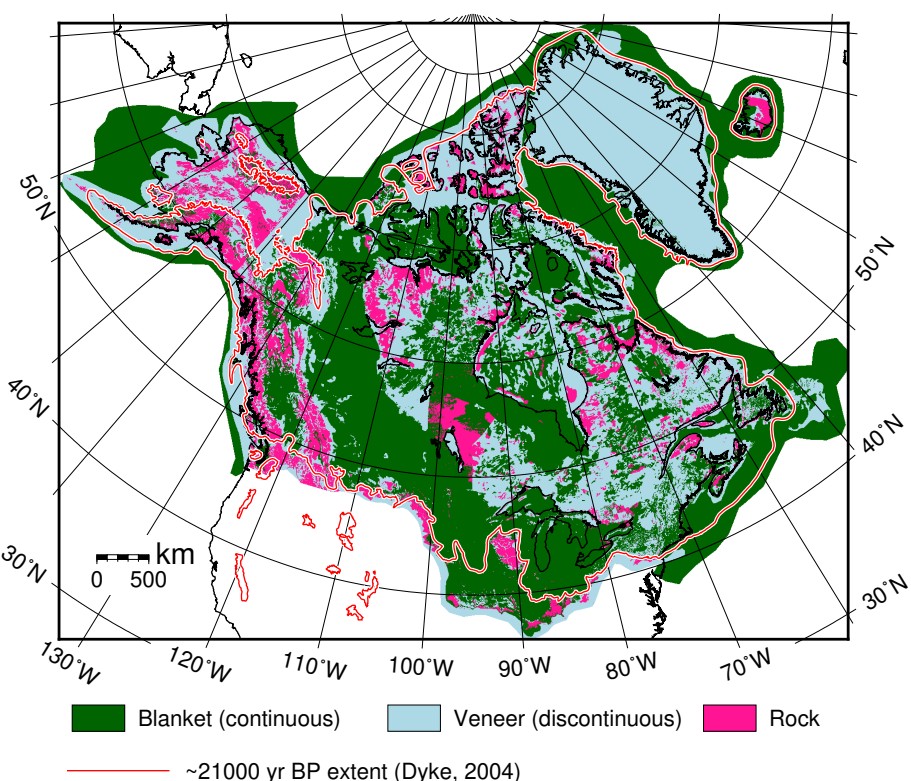

**Figure 1.** Sediment distribution. The red line is the glacial limits during the Last Glacial Maximum, 21 000 yr BP (thousands of years ago). Blanket regions are where unconsolidated sediments form a continuous surface, veneer regions have variable amounts of rock outcrops and discontinuous sediment cover, while rock areas have little or no sediment cover.

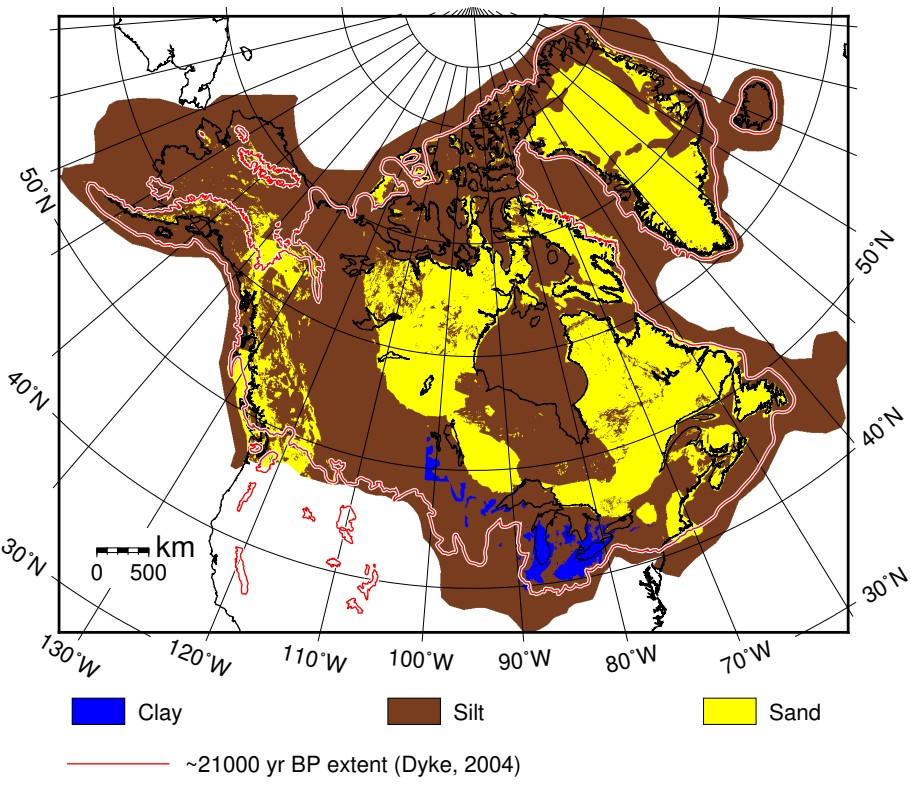

**Figure 2.** Sediment grain size. The red line is the glacial limits during the Last Glacial Maximum, 21 000 yr BP (Dyke, 2004). The types of sediment include clay (dominantly fine grained sediment), silt (an average composition between sand and clay), and sand (dominantly coarse grained sediments).



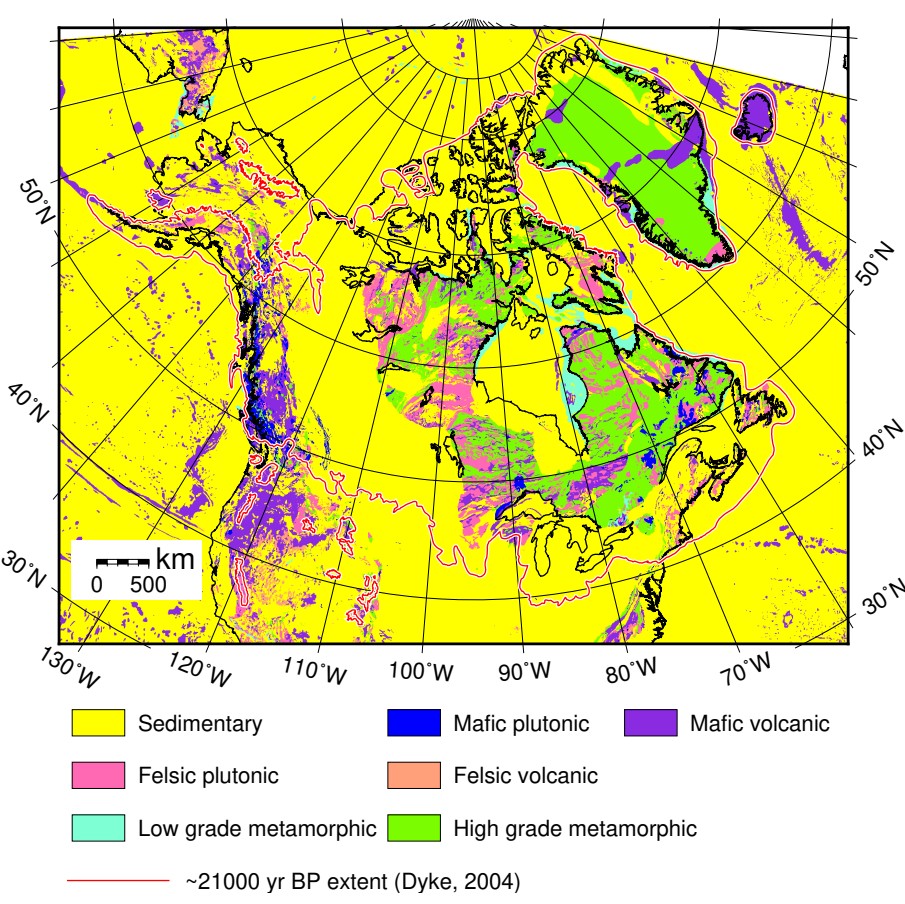

**Figure 3.** Generalized geology. The red line is the glacial limits during the Last Glacial Maximum, 21 000 yr BP (Dyke, 2004). The rock types are divided into sedimentary, felsic and mafic, volcanic and plutonic, and metamorphic categories.



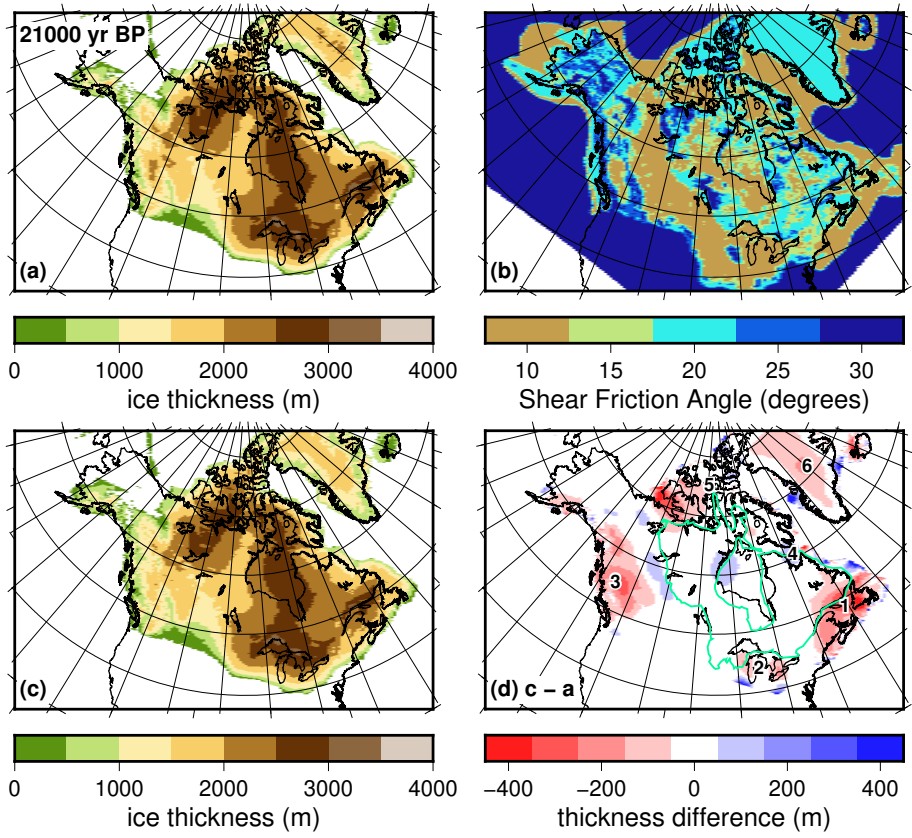

**Figure 4.** Impact of the distribution of sediments on the simulation of North American ice sheets. (a) Thickness of the ice sheets at 21 000 yr BP (after about 101 000 years of simulation) with the default shear friction angle, $\phi = 30°$. (b) Shear friction angle adjusted downwards for sediment cover. (c) Ice thickness at 21 000 yr BP using the shear friction angle shown in (b). (d) Difference in ice thickness between (a) and (c). The numbers in (d) represent areas mentioned in the text: (1) Eastern Canada, (2) Great Lakes, (3) Cordillera (4) Hudson Strait (5) Arctic Archipelago (6) Greenland. The green outline shows the exposed limit of the Canadian Shield.



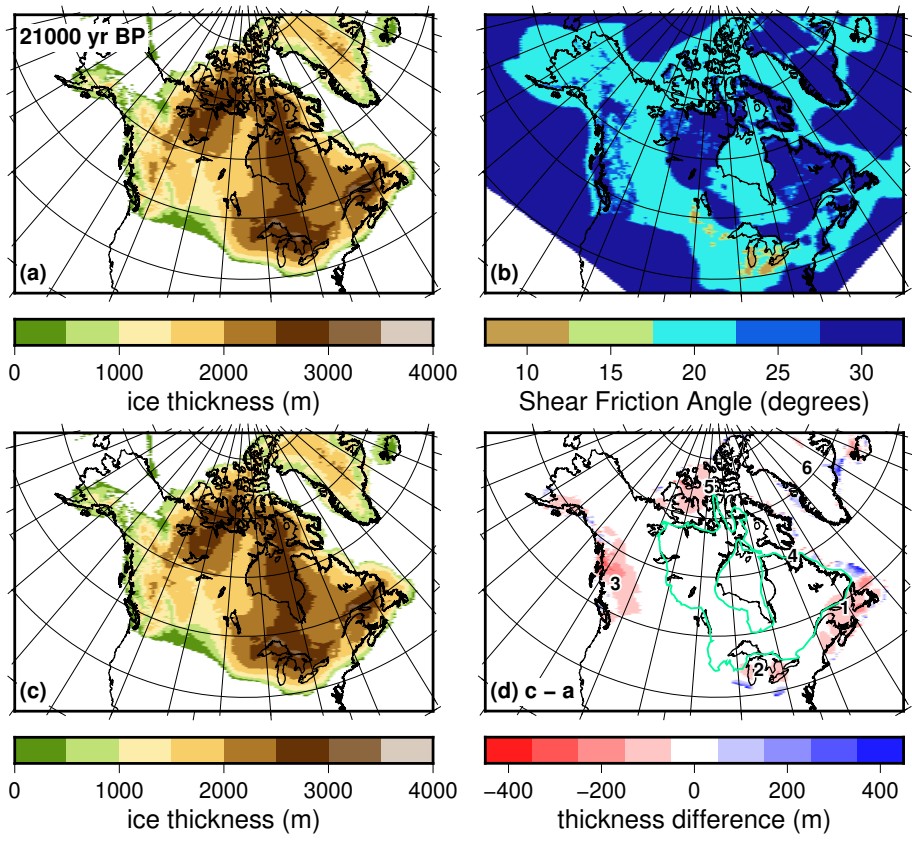

**Figure 5.** Impact of the grain size of sediments on the simulation of North American ice sheets. (a) Thickness of the ice sheets at 21 000 yr BP (after about 101 000 years of simulation) with the default shear friction angle, $\phi = 30°$. (b) Shear friction angle adjusted downwards for finer grained sediments. (c) Ice thickness at 21 000 yr BP using the shear friction angle shown in (b). (d) Difference in ice thickness between (a) and (c). The numbers in (d) represent areas mentioned in the text: (1) Eastern Canada, (2) Great Lakes, (3) Cordillera (4) Hudson Strait (5) Arctic Archipelago (6) Greenland. The green outline shows the exposed limit of the Canadian Shield.



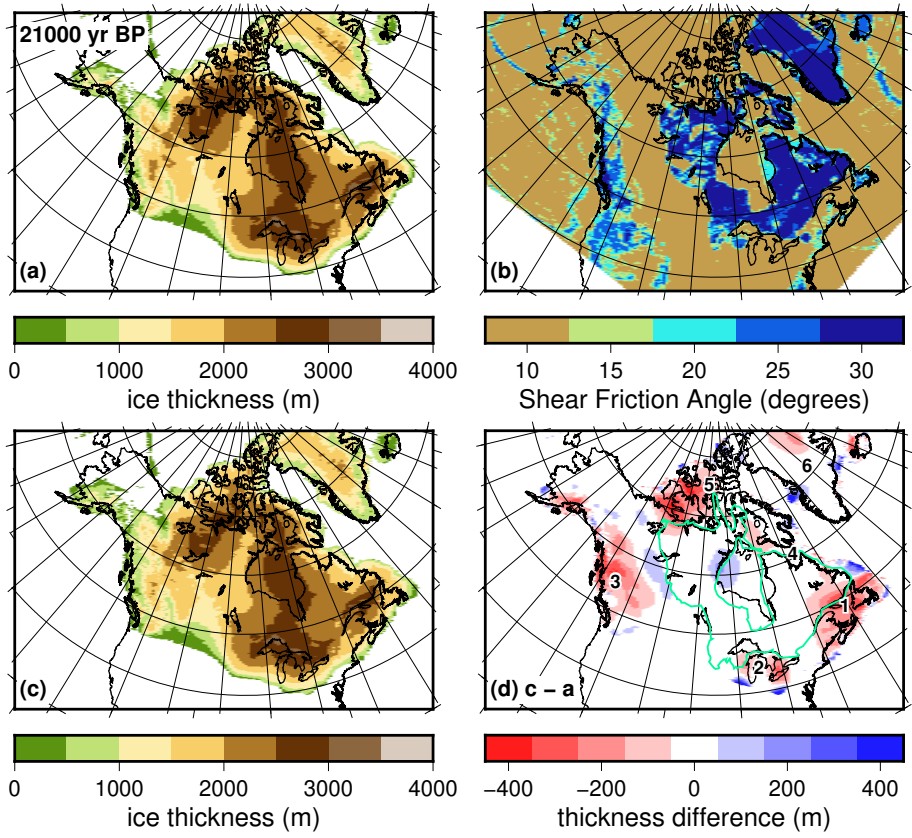

**Figure 6.** Impact of the geology on the simulation of North American ice sheets. (a) Thickness of the ice sheets at 21 000 yr BP (after about 101 000 years of simulation) with the default shear friction angle, $\phi = 30°$. (b) Shear friction angle adjusted downwards sediments and volcanic rocks. (c) Ice thickness at 21 000 yr BP using the shear friction angle shown in (b). (d) Difference in ice thickness between (a) and (c). The numbers in (d) represent areas mentioned in the text: (1) Eastern Canada, (2) Great Lakes, (3) Cordillera (4) Hudson Strait (5) Arctic Archipelago (6) Greenland. The green outline shows the exposed limit of the Canadian Shield.



**Table 1.** Maps used for the creation of the distribution and grain size dataset

| Map region | dataset used | reference |
|---|---|---|
| Canada | distribution | Fulton (1995); Geological Survey of Canada (2014) |
| Canada | grain size | Wheeler et al. (1996) |
| Continental United States west of the Rocky Mountains | distribution | Soller and Garrity (2018) |
| Ottawa Quadrangle | distribution, grain size | Fullerton et al. (1993) |
| Quebec Quadrangle | distribution, grain size | Borns et al. (1987) |
| Boston Quadrangle | distribution, grain size | Hartshorn et al. (1991) |
| Hudson River Quadrangle | distribution, grain size | Fullerton et al. (1992) |
| Sudbury Quadrangle | distribution, grain size | Sado et al. (1993) |
| Lake Erie Quadrangle | distribution, grain size | Fullerton et al. (1991) |
| Blue Ridge Quadrangle | distribution, grain size | Howard et al. (1991) |
| Lake Nipigon Quadrangle | distribution, grain size | Sado et al. (1994) |
| Lake Superior Quadrangle | distribution, grain size | Farrand et al. (1984) |
| Chicago Quadrangle | distribution, grain size | Lineback et al. (1983) |
| Louisville Quadrangle | distribution, grain size | Gray et al. (1991) |
| Lake of the Woods Quadrangle | distribution, grain size | Sado et al. (1995) |
| Minneapolis Quadrangle | distribution, grain size | Goebel et al. (1983) |
| Des Moines Quadrangle | distribution, grain size | Hallberg et al. (1994) |
| Ozark Plateau Quadrangle | distribution, grain size | Whitfield et al. (1993) |
| Winnipeg Quadrangle | distribution, grain size | Fullerton et al. (2000) |
| Dakotas Quadrangle | distribution, grain size | Fullerton et al. (1995) |
| Platte River Quadrangle | distribution, grain size | Swinehart et al. (1994) |
| Wichita Quadrangle | distribution, grain size | Denne et al. (1993) |
| Regina Quadrangle | distribution, grain size | Fullerton et al. (2007) |
| Montana | distribution, grain size | Fullerton et al. (2004, 2012, 2013, 2016) |
| Southern Cordillera Ice Sheet | distribution, grain size | Soller et al. (2009) |
| Nova Scotia | grain size | Stea et al. (1992) |
| Prince Edward Island | grain size | Prest (1973) |
| New Brunswick | grain size | Rampton (1988) |
| Newfoundland and Labrador | distribution, grain size | Government of Newfoundland and Labrador (2013) |
| Quebec | grain size | Thériault et al. (2012) |
| Northern Ontario | distribution, grain size | Ontario Geological Survey (1997) |
| Southern Ontario | distribution, grain size | Ontario Geological Survey (2003) |
| Manitoba | distribution, grain size | Matile and Keller (2006) |



| Map region | dataset used | reference |
| --- | --- | --- |
| Northern Saskatchewan | distribution, grain size | Schreiner (1984) |
| Alberta | grain size | Wheeler et al. (1996) |
| British Columbia | grain size | Massey et al. (2005) |
| Southwestern British Columbia | grain size | Clague et al. (1982) |
| Cordillera Ice Sheet | distribution | Eyles et al. (2018) |
| Yukon | distribution, grain size | Lipovsky and Bond (2014) |
| Yukon | grain size | Yukon Geological Survey (2016) |
| Alaska | distribution | Karlstrom (1964) |
| Alaska | grain size | Wilson et al. (2015) |
| Mainland Northwest Territories, Nunavut, and Baffin Island | grain size | Harrison et al. (2011) |
| Offshore Newfoundland and Grand Banks | distribution | King (2014) |
| Hudson Strait | distribution, grain size | MacLean (2001) |
| Gulf of St. Laurence | distribution, grain size | Loring and Nota (1973); Josenhans and Lehman (1999) |
| Labrador Shelf | distribution | Piper et al. (1990) |
| Southwestern Greenland | distribution | Weidick and Christoffersen (1974, 1978); Weidick and Klüver (1987) |
| Central eastern Greenland | distribution | Funder and Klüver (1988) |
| Southeastern Greenland and Iceland | distribution | Voges (1995) |
| Greenland | distribution | Sugden (1974) |
| Greenland and Iceland | grain size | Reed et al. (2004) |
| Greenland Ice Sheet | grain size | Dawes (2009) |



**Table 2.** Supplementary resources used for the creation of the distribution and grain size dataset

| Region | dataset used | notes | reference |
|---|---|---|---|
| Okanogan Lobe (southern Cordillera Ice Sheet) | distribution | Sediment cover is a veneer | Kovanen and Slaymaker (2004) |
| Puget Sound (southern Cordillera Ice Sheet) | distribution | The Puget Lobe overrode a thick sequence of proglacial sediments | Booth (1994) |
| Northern and Central Quebec | distribution, grain size | Dominantly sandy tills except in regions with sedimentary rocks | Bouchard (1989) |
| Ungava Peninsula, Quebec | distribution, grain size | Thick layers of coarse grained diamiction | Gray and Lauriol (1985) |
| Hudson Bay Lowlands, Ontario | grain size | Glacial sediments contain roughly equal amounts of clay, silt and sand | Thorleifson et al. (1992) |
| Southeastern Manitoba | grain size | Grain size of glacial sediments underneath Lake Agassiz deposits is dependent on the source region, but on average is silt | Teller and Fenton (1980) |
| Lake Winnipeg, Manitoba | distribution | Glacial sediments are discontinuous under the entire lake, except where there are end moraines. | Todd et al. (1998) |
| Alberta Interior Plains | grain size | Glacial sediments have a relatively uniform composition that is roughly equal parts clay, silt and sand. | Klassen (1989) |
| British Columbia | grain size | Glacial sediments generally have similar composition as underlying bedrock, though more coarse at higher elevations | Clague (1989) |
| British Columbia interior | grain size | Glacial sediments are silt or sand rich | Plouffe (2000) |
| Mainland Northwest Territories and Nunavut | grain size | Glacial sediments generally have grain size reflective of bedrock geology | McMartin et al. (2006) |
| Western Northwest Territories | grain size | Areas overlying the Western Canadian Sedimentary Basin have an unsorted mixture of sand silt and clay | *e.g.* Duk-Rodkin and Hughes (1993) |
| Hudson Bay | distribution | Multibeam data collected from Hudson Bay, which was ultimately used in Fulton (1995) | Josenhans and Zevenhuizen (1990) |
| Eastern Hudson Bay | distribution, grain size | Betcher Islands are relatively barren of unconsolidated sediments, bedrock is Proterozoic sedimentary and volcanic rocks | Jackson (2012) |
| St. Laurence estuary | distribution | Thick accumulations of glacial sediments only occur where there are bedrock troughs | Duchesne et al. (2010) |
| Offshore Nova Scotia | distribution | Seismic and multibeam data indicates significant glacial sediment accumulation | Todd et al. (1999); Todd and Shaw (2012) |
| Gulf of Maine | distribution, grain size | There is a thick succession of fine grained sediments, near Cape Cod it is more sandy | Uchupi and Bolmer (2008) |



| Region | dataset used | notes | reference |
|---|---|---|---|
| Northern Northwest Passage, Arctic Canada | distribution | Multibeam data indicates limited cover by glacial sediments | Niessen et al. (2010) |
| Gulf of Boothia | distribution | Multibeam data indicates of continuous layer of sediments | MacLean et al. (2010) |
| Coronation and Amundsen gulfs | distribution | Sediment veneer in the Coronation Gulf and inner Amundsen Gulf, thicker in the outer Amundsen Gulf | MacLean et al. (2015) |
| Western Lake Superior | distribution | Seismic data indicates that glacial sediment units are not continuous | Scholz (1984) |
| Western Lake Superior near Thunder Bay | distribution, grain size | Thick glacial sediment units that were interpreted to be fine grained | Gustafson (2012) |
| Lake Superior and Lake Michigan | distribution, grain size | Thick sediment cover with composition that reflects local geology for Lake Superior, and high clay content for Lake Michigan | Lineback et al. (1979) |
| Lake Ontario | distribution | The core of Lake Ontario has thick glacial sediment cover, but on the margins it is thin and discontinuous | Hutchinson et al. (1993); Lewis et al. (1995) |
| Lake Erie | distribution, grain size | Erie Lobe sediments are clay rich due to reworking of lake sediments | Karrow (1989) |
| Eastern Great Slave Lake | distribution | Glacial sediments are thick in some areas, but is not continuous | Christoffersen et al. (2008) |
| Beaufort Sea | distribution | Seismic data indicates thick sediment cover | Batchelor et al. (2013) |
| Greenland | distribution | Thick glacial sediment cover generally only exists in fjords and high plateaus | Funder (1989) |
| Greenland | distribution | Sediment cover in areas described by Sugden (1974) generally completely covers the bedrock | Corbett et al. (2015); Larsen et al. (2010); Håkansson et al. (2009) |
| Offshore Greenland | distribution | areas with scoured bedrock visible from multibean and seismic data have limited sediment cover, and smooth topography is more continuous | Funder (1989); Morlighem et al. (2017); Freire et al. (2015); Dowdeswell et al. (2010) |



| Region | dataset used | notes | reference |
|---|---|---|---|
| Greenland Ice Sheet | distribution | Seismic evidence indicates the presence of sediments under the ice sheet | Walter et al. (2014); Kulessa et al. (2017) |
| Greenland Ice Sheet | distribution, grain size | Most of Greenland is underlain by Archean and Paleo-proterozoic cratons, which are composed largely of high grade metamorphic and plutonic rocks, and likely has similar characteristics as the Canadian Shield. | St-Onge et al. (2009); Henriksen et al. (2009) |
| Offshore Iceland | distribution, grain size | Seismic surveys indicate thick sediment cover with relatively fine grain size | Principato et al. (2005) |
| Western Iceland | grain size | Older glacial sediments have been described as being silt rich and sandy-silt | Hjort et al. (1985); Ingólfsson (1985) |

**Table 3.** General properties of sediments relating to composition and texture

| Material | Grain Size (mm) | Shear Friction Angle | Cohesion | Permeability | Dilation |
|---|---|---|---|---|---|
| clay | <0.005 | <20 | >10 kPa | low | appreciable |
| silt | 0.05-0.005 | <30 | <10 kPa | variable | variable |
| sand | >0.05 | >30 | negligible | high | none |