# Peer review of "Geology datasets in North America, Greenland and surrounding"

_Earth System Science Data, 2018_

## Referee Comment (RC1) · Anonymous Referee #1 · 8 Dec 2018

General comments:

This paper presents three geologic maps spanning northern North America and Greenland, showing quantities related to sediment at the ice-bed interface. As discussed, sediment has important effects on basal sliding and hydrology in ice sheet models of Quaternary glacial cycles. The effect of these maps is illustrated using the PISM ice sheet model, comparing simulations at LGM to those using uniform sediment properties. The geological data sources, methods of assimilation and resulting maps are well described, although that is mostly outside my experience, and the points below are from an ice-sheet modeling perspective.

The datasets themselves (Figs. 1-3) may be a worthwhile contribution for this area of ice-sheet modeling. However, the relationship to model parameters and usage presented in the paper is not as useful. An alternative would be not to include any linkage at all with physics and modeling, and just to present Figs. 1 to 3 as descriptions of the real world, and leave it to users to make the links to ice-sheet models. But the former is included here (Table 3, Eq. (1) and the PISM runs), and its physical basis could be improved and made more useful, as discued below.

Specific comments:

1. A Coulomb rheology model is adopted as used in the PISM model (Eq. 1), and one parameter (shear friction angle phi) is chosen to be related to the geologic quantities mapped here. There is little discussion (section 3.1, pg. 7) on the physics of how the map properties in Figs. 1-3 are related to the model's phi, and why phi is chosen over other model parameters such as those involved with N for instance (in Eq. 1, involving basal hydrology). The values chosen for the range of phi (10, 20, 30 degrees, pg. 8 lines 28-29) are not justified well; a reference could be given for how those relate to extremes of hard bed vs. deformable sediment (see (5) below). More discussion would be helpful on how the 3 quantities mapped here (sediment distribution, grain size, bedrock type) relate physically to sliding, why three maps are useful rather than one, and how they relate to other parameters in sliding models and not just phi. As a bonus, relations to Weertman sliding coefficients could be provided as well as to Coulomb parameters.

2. In section 2.3 discussing sediment properties, the text says "Glacial sediments tend to be very poorly sorted, so these values should be assessed as being an average composition" (pg. 5 lines 12-13). Confusingly, the rest of the section seems to discuss grain-size properties only of the fine-grained material (matrix) that enclose pebbles and boulders (clast), and exclude the pebbles+boulders themselves. (Grain-size values in Table 3 are sub-millimeter). Do the clay and silt classes (pg. 5, lines 18-19) have significant fractions of pebbles+boulders, which is only mentioned for sand (line 21)? If so, what are the ranges of that fraction for each class? Do pebbles+boulders significantly influence the bulk(?) physical properties in Table 3? Both clast and matrix play cru-

cial roles in sediment deformation in Evans et al.'s comprehensive review (referenced here: Earth-Sci. Rev., 2006), especially their sections 5.1 to 5.4 on the "till-matrix framework".

3. Another perhaps naive physical question: if sedimentary till has been transported over long distances under the ice in previous glacial cycles, wouldn't the sediment properties be related to parent bedrock type(s) far upstream, and not to the local bedrock type as assumed here? This seems to be addressed by lines 26-28 on pg. 5 referring qualitatively to Fulton (1989), but isn't large-scale sediment transport a major feature of Quaternary glaciations?

4. (related to (1) above). The spatial patterns of the 3 datasets in Figs. 1-3 are all quite similar. That is, roughly speaking they all show mainly one value for Hudson Bay and the outer regions of the former LGM Laurentide, and mainly another value for Canadian Shield regions surrounding Hudson Bay, the Canadian Rockies, and Greenland. So not surprisingly the patterns of model LGM delta(ice-thickness) sensitivities shown in Figs. 4d-6d are all quite similar to each other.

Although the patterns are similar, the magnitudes of phi in Figs. 4b-6b differ. For instance, the values over Hudson Bay vs. Canadian Shield regions are generally ∼10 vs. 20 degrees in Fig. 4b using map # 1, ∼20 vs. 30 degrees in Fig. 5b using map # 2, and ∼10 vs. 30 degrees in Fig. 6b using map # 3. It is left for ice-sheet modelers to choose which map is best for phi, for which no guidance is given.

5. Model ice-thickness results in Figs. 4-6 are compared with those using a uniform value of phi=30. The delta(thickness) magnitudes in the (d) panels seem smaller and much less extensive than in early Laurentide studies exploring sediment vs. hard-bed effects, such as Fisher et al. (1985) and Licciardi et al. (1998) referenced here on pg. 2, line 14. Perhaps this is due to smaller contrasts in yield stress here than in the earlier studies, and a larger range of phi could be used than 10 to 30 degrees.

6. In several places the text mentions or implies stratigraphic relationships between the

underlying bedrock, layers of glacially derived sediment, perhaps layers of earlier (or Holocene?) non-glacial sediment, and water bodies; for instance on pg. 3 lines 1-3, pg. 4 lines 1-4, pg. 4 line 19. For a non-geologist, it is challenging to conceptualize this clearly, and to keep straight what the stratigraphic relationships are (for glacial vs. non-glacial sediment), and whether they are important. I suggest slightly expanded discussion in places. Also a simple diagram would be helpful that shows the various idealized stratigraphic cases, with labeled horizontal rectangles one on top of each other. The cases would always have underlying bedrock, then one case with bedrock covered by glacial sediment (perhaps with horizontal breaks indicating patchiness), another with bedrock + non-glacial sediment, another with bedrock + glacial sediment + water body, etc.

7. If possible, ranges of thickness values for each type of layer could be shown in the diagram suggested above. These are mentioned in places in the text, and could be gathered there. The paper says (pg. 7 line 14) "it is not possible to give a detailed quantitative estimate of (sediment) distribution and thickness", but even rough ranges would be helpful. Melanson et al. (QSR, 2013, Fig. 1), and Hildes et al. (QSR, 2004, Fig. 4) produced sediment thickness maps based on Fulton et al. (2005) referenced here; could a guide be provided here? Maps of sediment thickness are not so important for models of LGM or the last glacial cycle, but are more helpful for longer-term studies of Plio-Pleistocene variations involving sediment evolution, such as Ganopolski et al., Clim. Past, 2011.

8. The areas in Figs. 4d-6d where model LGM delta(ice thicknesses) are appreciable occur mostly around the margins of the ice sheet. The paper suggests this is controlled in part by areas where the bed is sufficiently warm and wet to allow sliding in the PISM model (pg. 8 line 35). This could be corroborated by showing a model map of basal temperature or basal melt rate for 21 ka.

Technical points:

pg. 5 lines 30-33: The classification relating bedrock to grain size is given for silt (line 30) and sand (line 32), but seems to be missing for clay.

pg. 6 line 11: The name "Felsic volcanic" is used twice. Possibly the 2nd should be "Mafic volcanic".

pg. 7 line 4-6: Table 3 is discussed in section 3.1, before Eq. (1) and shear friction angle and cohesion (quantities in the table) are discussed in section 3.2. This could be resolved easily by referring to Eq. (1) and following text in the table caption.

pg. 7 lines 9-11. The two consecutive sentences beginning "The patchiness of sediment..." and "The lack of sediment.." seem to say nearly the same thing. Is the distinction one of horizontal scale?

pg. 7 line 17: Regarding drainage of water under the ice, perhaps clarify in a few words that the permeability of bedrock under the sediment "aquifer" is involved here, I think. (As opposed to permeability of sediment as in Table 3, which would use map # 2 for grain size and not map # 3 for bedrock type).

pg. 8 line 34; pg. 9 line 13; pg. 9 line 22: These sentences say that south of the Great Lakes at LGM, the ice sheet extends further than in the reference simulation. However there is no discernible difference in the margin locations south of the Great Lakes in Figs. 4a vs. c, 5a vs. c, 6a vs. c. How many km does "further south" mean in the sentences, and why is it not visible in the figures?

Minor corrections/suggestions: pg. 1 line 4: Should be "distribution of surficial". pg. 3 line 22: "When" should be "In". pg. 4 line 11: Is "bend simplify" intended? pg. 4 line 16; pg 5 line 9; pg. 7 line 4; pg. 9 line 9,15: "on" should be "in". pg. 6 line 18: Should be "as most of these". pg. 7 line 11: Should be "of the latter". pg. 7 line 17: "later" should be "latter".

---

## Referee Comment (RC2) · Ely (Referee) · 9 Jan 2019

General comments: This paper presents useful datasets regarding the geology and sedimentology of North America, Greenland, Iceland and parts of Russia. The datasets are extensive and useful for the community. The demonstration of use in ice sheet models is interesting and also useful for the reader to consider how future experiments with this and similar datasets might be conducted. However, the aim of these simulations for this paper should be noted in the text (i.e. to demonstrate utility of the dataset, not to draw scientific conclusions about the Laurentide which I presume is the focus of a later paper). The authors should be applauded for citing all the original literature that goes into this dataset in Tables 1 and 2. It may also be useful to distribute a similar bibliography with the dataset. This is therefore a very worthwhile contribution

to the literature and I hope that my comments below can help improve the manuscript.

Specific comments: The title - only North America is mentioned, yet the data includes Greenland and Iceland. I think these places need incorporating in the title somehow. As far as I can tell from figures (could not check the actual data as pangea is password protected) the whole of North America is not covered by the data either.

Uncertainty. In creating the dataset the authors necessarily and reasonably had to make some interpretations and interpolations between sources of data. However, there doesn't seem to be any record of where gaps have had to be filled or boreholes consulted. A map of data coverage (boreholes, geology map location etc), or another map showing some sort of confidence level in the data would be very useful for those using the data in model experiments and for focusing future work.

Some notes on the data format that the data is provided in would be useful. As mentioned above, I could not check as not yet accessible through the repository.

Minor comments: Abstract, page 1, line 5. Please state the scale or resolution of the dataset.

P1, L 17 - I find the use of the word substrate odd here, it is more about whether sediments were present beneath the ice sheet or not. To me, these sediments (or bedrock) would then be the substrate, i.e. the bit in contact with the bed of the ice is the substrate.

P1, L 19 - Consider adding the more recent reference of Storrar et al., 2014 on eskers across the Laurentide.

P2, L 9 - Unclear subject matter. Presence or absence of what?

P2, L 15-17 - This section needs better linking to scope of the paper. These are all important factors, but some better crafting of the paragraph is required to state why we need to know about these things. In particular, here the subject jumps from the Laurentide to Svalbard without any linking.

P3, L 6-9 - These sentences are better incorporated into the following section.

P3, L 13 - use of word "extended" is technically correct, but I wonder if better for reader if you use occurred between or similar, given the use of the word "extent" later to refer to where the ice got to.

P3, L 32 - requires rewording. Perhaps "information" rather than "glacial"

Section 2.1. - A statement on the intended use and resolution of the data would be useful for those intending to use the dataset and to prevent misuse. I imagine the datasets will be useful for those doing ice sheet-scale experiments. However, the resolution may limit utility for those interested in a single outlet glacier/ice stream for example.

P5, L 30 - No notes on clay

P6, Section 2.5. This section would be useful for including the notes/map of "uncertainty" stated above.

P7, L 23. I think it worth restating here for the audience that your aim is not to draw specific conclusions about the form of the Laurentide in this paper. The following sections (3.2.1 to 3.2.3) do mention specifics of the modelled ice sheet. However, I think that these are safe as they fall short of evaluating whether there is an improvement or not, by just stating that there is a change induced by the data.

Additional references: Storrar, R.D., Stokes, C.R. and Evans, D.J., 2014. Morphometry and pattern of a large sample (> 20,000) of Canadian eskers and implications for subglacial drainage beneath ice sheets. Quaternary Science Reviews, 105, pp.1-25.

---

## Author Comment (AC1) · 11 Jan 2019

Dr. Ely pointed out that there was a password on the Pangaea dataset. I did not realize that this was the case, and I have had the password removed. The dataset should now be accessible.

---

## Author Comment (AC2)

**Response to Anonymous referee #1**

Evan J. Gowan[1], Lu Niu[1], Gregor Knorr[1], and Gerrit Lohmann[1]

[1]Alfred Wegener Institute, Helmholtz Centre for Polar and Marine Research, Bremerhaven, Germany

**Correspondence:** Evan J. Gowan (evan.gowan@awi.de)

**1   Introduction**

We would like to thank the reviewer for his or her or their comments. We have responded to the comments below. The comments from the reviewer is in italics, while our response is in normal font.

**2   General comments**

5   *This paper presents three geologic maps spanning northern North America and Greenland, showing quantities related to sediment at the ice-bed interface. As discussed, sediment has important effects on basal sliding and hydrology in ice sheet models of Quaternary glacial cycles. The effect of these maps is illustrated using the PISM ice sheet model, comparing simulations at LGM to those using uniform sediment properties. The geological data sources, methods of assimilation and resulting maps are well described, although that is mostly outside my experience, and the points below are from an ice-sheet modeling perspective.*

10   *The datasets themselves (Figs. 1-3) may be a worthwhile contribution for this area of ice-sheet modeling. However, the relationship to model parameters and usage presented in the paper is not as useful. An alternative would be not to include any linkage at all with physics and modeling, and just to present Figs. 1 to 3 as descriptions of the real world, and leave it to users to make the links to ice-sheet models. But the former is included here (Table 3, Eq. (1) and the PISM runs), and its physical basis could be improved and made more useful, as discused below.*

15   In creating this paper, we had discussions about the appropriateness of including some ice sheet simulations using the datasets in this paper. The target of this journal is purely to describe the datasets, which could then be used by ice sheet modellers to improve their ice sheet simulations. The lead author (Evan J. Gowan) decided to include the simulations with the purpose of demonstrating "if you change the basal conditions on the basis of these maps, there will be an impact on resulting ice sheet, so they should be considered". We feel a more thorough analysis on the parameters that should be chosen is beyond the

20   scope of the paper. The default PISM model does not have the capabilities to implement the various parameters described in the paper (namely, discontinuous sediment cover), and were only included here as a guide as to what could be implemented into the ice sheet model if the maps presented here are to be used. The values of phi, with the except of the sediment grain

size maps, were selected only to show contrasts between the different map units, on the basis of expected impact on the ice sheet, and do not have any physical meaning. The sediment distribution, for instance, should be parameterized in a way that sediment deformation no longer has as much impact on the ice sheet dynamics. In PISM, there is no way to do this (although as mentioned late in the paper, we are in the process of incorporating this).

5    Upon reading the ice sheet modelling section, we realize that we were not explicit enough with the purpose of the ice sheet modelling (since both the reviewers have brought this up). Although we attempted to articulate this in third paragraph of section 3.2, we have rewritten the introductory paragraph in section 3.2 to clarify our intentions:

> To show the utility of the dataset, we incorporate the information for use with the ice sheet model PISM 1.0 (Bueler and Brown, 2009; PISM authors, 2017), with the addition of an index forcing scheme described in Niu et al. (2017).
> 10  In the standard version of PISM, the model for basal sliding has an assumption that there is a continuous layer of sediments underlying the ice sheet. Obviously, in areas where sediment coverage is discontinuous, this is not a valid model. Therefore the purpose of the following simulations are simply to demonstrate that if there is a contrast in the basal conditions based on the underlying geological parameters, there will be an impact on the resulting ice sheet simulation. The simulations are not necessarily reflective of actual basal conditions of the ice sheet.

**15  3  Specific comments**

*1. A Coulomb rheology model is adopted as used in the PISM model (Eq. 1), and one parameter (shear friction angle phi) is chosen to be related to the geologic quantities mapped here. There is little discussion (section 3.1, pg. 7) on the physics of how the map properties in Figs. 1-3 are related to the model's phi, and why phi is chosen over other model parameters such as those involved with N for instance (in Eq. 1, involving basal hydrology). The values chosen for the range of phi (10, 20,*
20  *30 degrees, pg. 8 lines 28-29) are not justified well; a reference could be given for how those relate to extremes of hard bed vs. deformable sediment (see (5) below). More discussion would be helpful on how the 3 quantities mapped here (sediment distribution, grain size, bedrock type) relate physically to sliding, why three maps are useful rather than one, and how they relate to other parameters in sliding models and not just phi. As a bonus, relations to Weertman sliding coefficients could be provided as well as to Coulomb parameters.*

25  As mentioned above, the Coulomb rheology model that we use in these simulations is not realistic to describe the actual basal conditions of the ice sheet. It does not truly describe a hard versus soft bed situation, since it does not include parameters on how sediment distribution affects ice sheet flow (as mentioned in paragraph 2 in section 3.1), or how bedrock could drain water underneath the ice sheet (as mentioned in paragraph 3 of section 3.1). As mentioned above, these simulations are only to illustrate that if you change the basal conditions on the basis of these maps, there is an impact on the ice sheet.

For the sediment distribution maps, we justified the parameters (see lines 28-30 on page 8) by setting "hard bedded" regions with the rock classification to have a high shear angle (which increases the strength of the base), and a low value where there is continuous sediments where it would be suspected to be weak.

The sediment grain size dataset used values that were based on measured values of the shear friction angle (lines 6-8 on page 9)

In the geology dataset, we changed the friction angle based on how easily the bedrock can erode to produce glacial sediments (lines 17-19 on page 9).

We do not wish to comment on Weertman parameters, as it is not clear that the Weertman sliding law is valid (Fowler, 2010), and there are few observations of what the bed roughness parameter is to attempt such a discussion (see section 7.2.2.1 in Cuffey and Paterson (2010)). The ice sheet model we use (PISM), also does not include a Weertman sliding law.

*2. In section 2.3 discussing sediment properties, the text says "Glacial sediments tend to be very poorly sorted, so these values should be assessed as being an average composition" (pg. 5 lines 12-13). Confusingly, the rest of the section seems to discuss grain-size properties only of the fine-grained material (matrix) that enclose pebbles and boulders (clast), and exclude the pebbles+boulders themselves. (Grain-size values in Table 3 are sub-millimeter). Do the clay and silt classes (pg. 5, lines 18-19) have significant fractions of pebbles+boulders, which is only mentioned for sand (line 21)? If so, what are the ranges of that fraction for each class? Do pebbles+boulders significantly influence the bulk(?) physical properties in Table 3? Both clast and matrix play crucial roles in sediment deformation in Evans et al.'s comprehensive review (referenced here: Earth-Sci. Rev., 2006), especially their sections 5.1 to 5.4 on the "till-matrix framework".*

In retrospect, we agree that it is confusing that the discussion only refers to the fine components of the glacial sediment. Glacial sediment has a bimodal grain size distribution, no glacial geology map used in this compilation describes the coarse fraction. We agree, having a quantitative assessment of the fraction of the till that is coarser than sand size would be a useful parameter to know, as having more coarse material will make the till stronger. Quantitative measurements simply do not exist in most areas. The qualitative classifications are also dependent on the author/organization, which is why we simplified the dataset to just three unit types. We have revised the first paragraph of section 3.2 to emphasize that that grain size is referring only to the fine component of the sediment.

The map of generalized grain size of glacial sediments is shown on Figure 5. A glacial sediment, diamiction or till (the later has a definitive glacial origin) is an unsorted material with grain size ranging from clay to boulder. Glacial sediments generally have a bimodal grain size distribution, with peaks in the course (pebble to boulder) and fine (clay to sand) fractions (Dreimanis and Vagners, 1971). The relative amount of course to fine is dependent on the distance from the source of the course material, so on glacial geology maps and datasets, glacial sediments are described in terms of the fine fraction only. To simplify the classification, we only have three main classification types, based on the dominant grain size of fine fraction. This classification scheme is based on the Surficial Materials in the Conterminous United States map (Soller and Reheis, 2004), and we attempted to unify this scheme

with maps and data in Canada. The grain size of the sediments tends to have geographical dependence. As an example, in the map by Soller and Reheis (2004), clay rich glacial sediment exists in areas around the Great Lakes, where source material was derived from lake sediments, and sandy glacial in mountainous regions where there are extensive rock outcrops. The relative fraction of the sediment that is coarser than sand is not possible to quantify, since most of the data sources only give qualitative descriptions of the coarse fraction.

*3. Another perhaps naive physical question: if sedimentary till has been transported over long distances under the ice in previous glacial cycles, wouldn't the sediment properties be related to parent bedrock type(s) far upstream, and not to the local bedrock type as assumed here? This seems to be addressed by lines 26-28 on pg. 5 referring qualitatively to Fulton (1989), but isn't large-scale sediment transport a major feature of Quaternary glaciations?*

Although it seems intuitive that glacial transport will strongly influence sediment composition, it actually isn't as large as one might think. For example, some of the lobes in the southern Laurentide region were sampled for zircons by Kassab et al. (2017), and they found that even though there was ice streaming from the Canadian Shield, the age of the zircons indicated that the source of the till still largely reflected the underlying Michigan Basin rocks. At any rate, the target of this paper is to represent the sediments for the last glaciation, in previous glaciations it was likely different.

*4. (related to (1) above). The spatial patterns of the 3 datasets in Figs. 1-3 are all quite similar. That is, roughly speaking they all show mainly one value for Hudson Bay and the outer regions of the former LGM Laurentide, and mainly another value for Canadian Shield regions surrounding Hudson Bay, the Canadian Rockies, and Greenland. So not surprisingly the patterns of model LGM delta(ice-thickness) sensitivities shown in Figs. 4d-6d are all quite similar to each other.*

*Although the patterns are similar, the magnitudes of phi in Figs. 4b-6b differ. For instance, the values over Hudson Bay vs. Canadian Shield regions are generally ~10 vs. 20 degrees in Fig. 4b using map #1, ~20 vs. 30 degrees in Fig. 5b using map #2, and ~10 vs. 30 degrees in Fig. 6b using map #3. It is left for ice-sheet modelers to choose which map is best for phi, for which no guidance is given.*

For the glacial sediment composition, the values of phi are probably well justified as it is based on actual values from (Cuffey and Paterson, 2010) (as noted in the text). For the rest of the properties, a completely different basal condition model should be used, as changes in basal conditions due to sediment distribution (for instance) are not based the till friction angle, but rather changes in the basal hydrological system. We are working on this (as noted on lines 14 and 15 on page 8 of the original manuscript). The experiments shown here are simply to show in a qualitative sense that changing the basal conditions on the basis of the maps affects the evolution of the ice sheet (as addressed above).

*5. Model ice-thickness results in Figs. 4-6 are compared with those using a uniform value of phi=30. The delta(thickness) magnitudes in the (d) panels seem smaller and much less extensive than in early Laurentide studies exploring sediment vs. hard-bed effects, such as Fisher et al. (1985) and Licciardi et al. (1998) referenced here on pg. 2, line 14. Perhaps this is due to smaller contrasts in yield stress here than in the earlier studies, and a larger range of phi could be used than 10 to 30 degrees.*

As noted on lines 9-13 on page 8 of the original manuscript, the impact on the ice sheet is limited only to areas where there is sufficient ice flow to cause melting at the base, and where that is not true, changing the shear friction angle has no effect. The figure added below for point #8 shows this. There is little change over most of the ice sheet because there is not enough water produced in the model to saturate the sediments. Further changing the shear friction angle in those regions will have no effect on the outcome. The range of shear friction angles used here fall within the range of realistic values for till, no geological material will have shear friction angles less than $10°$. This is why we are working on creating a new basal conditions model to better incorporate the geological parameters in a realistic way (as noted on lines 14 and 15 on page 8).

*6. In several places the text mentions or implies stratigraphic relationships between the underlying bedrock, layers of glacially derived sediment, perhaps layers of earlier (or Holocene?) non-glacial sediment, and water bodies; for instance on pg. 3 lines 1-3, pg. 4 lines 1-4, pg. 4 line 19. For a non-geologist, it is challenging to conceptualize this clearly, and to keep straight what the stratigraphic relationships are (for glacial vs. non-glacial sediment), and whether they are important. I suggest slightly expanded discussion in places. Also a simple diagram would be helpful that shows the various idealized stratigraphic cases, with labeled horizontal rectangles one on top of each other. The cases would always have underlying bedrock, then one case with bedrock covered by glacial sediment (perhaps with horizontal breaks indicating patchiness), another with bedrock + non-glacial sediment, another with bedrock + glacial sediment + water body, etc.*

We have added the following illustration showing the relationship between bedrock, glacial sediments, water bodies, and post-glacial sediments:

[Figure]

**Figure 1.** Illustration showing the relationship between the bedrock, glacial sediments and postglacial sediments. In glacial times, the ice sheet is in contact with glacial sediments created by the ice sheet itself, and bedrock. In post-glacial times, the bedrock and glacial sediments can be obscured by water bodies and post-glacial sediments.

We have added the following figure which shows the relationship between the sediment and bedrock in the distribution classes:

[Figure]

**Figure 2.** Illustration of how the sediment distribution relates to the underlying bedrock and thickness of the sediments. The rock class has only isolated patches of sediment, the veneer class has a thin sediment layer with bedrock outcrops and a visible influence of bedrock topography on the surface, while with the blanket class, the sediments completely obscure the bedrock surface.

*7. If possible, ranges of thickness values for each type of layer could be shown in the diagram suggested above. These are mentioned in places in the text, and could be gathered there. The paper says (pg. 7 line 14) "it is not possible to give a detailed quantitative estimate of (sediment) distribution and thickness", but even rough ranges would be helpful. Melanson et al. (QSR, 2013, Fig. 1), and Hildes et al. (QSR, 2004, Fig. 4) produced sediment thickness maps based on Fulton et al. (2005) referenced here; could a guide be provided here? Maps of sediment thickness are not so important for models of LGM or the last glacial cycle, but are more helpful for longer-term studies of Plio-Pleistocene variations involving sediment evolution, such as Ganopolski et al., Clim. Past, 2011.*

For the distribution dataset, the important parameter is not necessarily sediment thickness (although we have included some quantitative values in the description of the units), but rather the percentage of surface is covered in sediments versus bare rock. It would be difficult to put hard numbers on this, since many surficial geology maps only give qualitative categories, and the definition also varies depending on the author of the map (as mentioned in the text of section 2.2). To emphasize this point, we have added the following to the intro paragraph of section 2.2:

Many maps used in this dataset only give qualitative descriptions of the distribution, and the definition often varies between mappers. As a result, it is not possible to give an exact range for sediment thickness or percentage sediment cover. We recommend modellers explore a range of values.

The classification scheme used by Melanson et al. (2013) and Hildes et al. (2004) for defining the distribution of sediments is essentially the same as ours, which is to say it is a qualitative assessment of sediment distribution. We have added the following to section 2.2:

> A scheme similar to this has been used in the studies by Hildes et al. (2004) and Melanson et al. (2013) for use in the modelling of sediment transport. The difference in our dataset is that we explicitly do not include post-glacial sediments, and instead try to fill these gaps with supplemental information.

As for thickness maps, they produced these using an empirical formula based on those same qualitative descriptors. In order to get a quantitative assessment of sediment cover in longer-term studies, it would be necessary to first to have data on the amount of sediment has actually been transported glacially. From that, and inverse model could perhaps be applied. Such a dataset does not currently exist, but from some discussions the lead author (Evan Gowan) has had with geologists, such a dataset is under construction for some areas in the United States.

*8. The areas in Figs. 4d-6d where model LGM delta(ice thicknesses) are appreciable occur mostly around the margins of the ice sheet. The paper suggests this is controlled in part by areas where the bed is sufficiently warm and wet to allow sliding in the PISM model (pg. 8 line 35). This could be corroborated by showing a model map of basal temperature or basal melt rate for 21 ka.*

We added the following plot that shows the basal heating.

[Figure]

**Figure 3.** Areas in the default simulation where basal frictional heating exceeds 0.01 W/m$^2$ (shown in red). The grey region is where there is grounded ice.

**4    Technical points:**

*pg. 5 lines 30-33: The classification relating bedrock to grain size is given for silt (line 30) and sand (line 32), but seems to be missing for clay.*

We did not include a clay unit here, because as mentioned on lines 14-16 on page 5, clay rich glacial sediments are likely derived from lake sediment, rather than bedrock.

*pg. 6 line 11: The name "Felsic volcanic" is used twice. Possibly the 2nd should be "Mafic volcanic".*

This has been fixed.

*pg. 7 line 4-6: Table 3 is discussed in section 3.1, before Eq. (1) and shear friction angle and cohesion (quantities in the table) are discussed in section 3.2. This could be resolved easily by referring to Eq. (1) and following text in the table caption.*

We added a reference to equation 1 to the table 3 title.

*pg. 7 lines 9-11. The two consecutive sentences beginning "The patchiness of sediment..." and "The lack of sediment.." seem to say nearly the same thing. Is the distinction one of horizontal scale?*

These two sentences refer to two different things. The first means that bedrock sticking out will resist flow due to increased friction, as the rock sticks into the glacier. The second sentence refers to the fact that the lack of sediments means there is no sediment deformation as a mechanism to flow. We modified the second sentence to make that clear.

> The lack of sediment in an otherwise sediment covered region may increase resistance to flow as well if sediment deformation is a dominant factor in controlling flow

*pg. 7 line 17: Regarding drainage of water under the ice, perhaps clarify in a few words that the permeability of bedrock under the sediment "aquifer" is involved here, I think. (As opposed to permeability of sediment as in Table 3, which would use map #2 for grain size and not map #3 for bedrock type).*

We appended "into the bedrock aquifer" at the end of the sentence.

*pg. 8 line 34; pg. 9 line 13; pg. 9 line 22: These sentences say that south of the Great Lakes at LGM, the ice sheet extends further than in the reference simulation. However there is no discernible difference in the margin locations south of the Great Lakes in Figs. 4a vs. c, 5a vs. c, 6a vs. c. How many km does "further south" mean in the sentences, and why is it not visible in the figures?*

We admit that you have to look very closely to identify that it indeed went further south. It is only one grid cell point, which is 20 km, which is visible by the blue area on the difference map. We have noted this in the text.

**5   Minor corrections/suggestions**

*pg. 1 line 4: Should be "distribution of surficial".*

Fixed.

*pg. 3 line 22: "When" should be "In".*

Using the word "when" is intentional.

*pg. 4 line 11: Is "bend simplify" intended?*

That is the name of the tool we used in ArcGIS.

*pg. 4 line 16; pg 5 line 9; pg. 7 line 4; pg. 9 line 9,15: "on" should be "in".*

Unless Copernicus' style guide requires this convention (and it doesn't appear to), we are leaving this as is.

*pg. 6 line 18: Should be "as most of these".*

Fixed

*pg. 7 line 11: Should be "of the latter". pg. 7 line 17: "later" should be "latter".*

Fixed.

**References**

Bueler, E. and Brown, J.: Shallow shelf approximation as a "sliding law" in a thermodynamically-coupled ice sheet model, J. Geophys. Res., 114, https://doi.org/10.1029/2008JF001179, 2009.

Cuffey, K. M. and Paterson, W. S. B.: The physics of glaciers, Elsevier, 2010.

Dreimanis, A. and Vagners, U. J.: Bimodal distribution of rock and mineral fragments in basal tills, in: Till, a symposium, edited by Goldthwait, R. P., pp. 237–250, Ohio State University Press, Columbus, Ohio, 1971.

Fowler, A.: Weertman, Lliboutry and the development of sliding theory, Journal of Glaciology, 56, 965–972, https://doi.org/10.3189/002214311796406112, 2010.

Hildes, D. H., Clarke, G. K., Flowers, G. E., and Marshall, S. J.: Subglacial erosion and englacial sediment transport modelled for North American ice sheets, Quaternary Science Reviews, 23, 409–430, https://doi.org/10.1016/j.quascirev.2003.06.005, 2004.

Kassab, C. M., Brickles, S. L., Licht, K. J., and Monaghan, G. W.: Exploring the use of zircon geochronology as an indicator of Laurentide Ice Sheet till provenance, Indiana, USA, Quaternary Research, 88, 525–536, https://doi.org/10.1017/qua.2017.71, 2017.

Melanson, A., Bell, T., and Tarasov, L.: Numerical modelling of subglacial erosion and sediment transport and its application to the North American ice sheets over the Last Glacial cycle, Quaternary Science Reviews, 68, 154–174, https://doi.org/10.1016/j.quascirev.2013.02.017, 2013.

Niu, L., Lohmann, G., Hinck, S., and Gowan, E. J.: Sensitivity of atmospheric forcing on Northern Hemisphere ice sheets during the last glacial-interglacial cycle using outputs from PMIP3, Climate of the Past Discussion, https://doi.org/10.5194/cp-2017-105, in review, 2017.

PISM authors: PISM, a Parallel Ice Sheet Model, http://www.pism-docs.org, 2017.

Soller, D. R. and Reheis, M. C.: Surficial Materials in the Conterminous United States, U.S. Geological Survey Open File Report OFR-03-275, U.S. Geological Survey, https://pubs.er.usgs.gov/publication/ofr2003275, scale 1:5,000,000, 2004.

---

## Author Comment (AC3)

**Response to Jeremy Ely (reviewer #2)**

Evan J. Gowan1, Lu Niu1, Gregor Knorr1, and Gerrit Lohmann1 1Alfred Wegener Institute, Helmholtz Centre for Polar and Marine Research, Bremerhaven, Germany **Correspondence:** Evan J. Gowan (evan.gowan@awi.de)

**1 Introduction**

We would like to thank Dr. Ely for his comments. We have responded to the comments below. The comments from the reviewer is in italics, while our response is in normal font.

**2 General comments**

- 5 General comments: This paper presents useful datasets regarding the geology and sedimentology of North America, Greenland, Iceland and parts of Russia. The datasets are extensive and useful for the community. The demonstration of use in ice sheet models is interesting and also useful for the reader to consider how future experiments with this and similar datasets might be conducted. However, the aim of these simulations for this paper should be noted in the text (i.e. to demonstrate utility of the dataset, not to draw scientific conclusions about the Laurentide which I presume is the focus of a later paper). The authors
- 10 should be applauded for citing all the original literature that goes into this dataset in Tables 1 and 2. It may also be useful to distribute a similar bibliography with the dataset. This is therefore a very worthwhile contribution to the literature and I hope that my comments below can help improve the manuscript.

As mentioned in the response to reviewer #1, the intention of our simulations was just as mentioned here - to demonstrate the utility of the datasets. We have updated the section to make this more clear (see response to reviewer #1 for details). Adding the bibliography to the datasets is a good idea, we have added it to the dataset.

**3** Specific comments**

15

The title - only North America is mentioned, yet the data includes Greenland and Iceland. I think these places need incorporating in the title somehow. As far as I can tell from figures (could not check the actual data as pangea is password protected) the whole of North America is not covered by the data either. We apologize for the inaccessibility of the datasets - Pangaea added a password for some reason. Once this was pointed out, we had the password removed.

This compilation started off as being purely focused on North America, the additions of Greenland and Iceland to the dataset came very late in the process. We will change the title to the following:

5 Geology datasets in North America, Greenland and surrounding areas for use with ice sheet models

follows.

20

25

2. In section 2.3 discussing sediment properties, the text says "Glacial sediments tend to be very poorly sorted, so these values should be assessed as being an average composition" (pg. 5 lines 12-13). Confusingly, the rest of the section seems to discuss grain-size properties only of the fine-grained material (matrix) that enclose pebbles and boulders (clast), and exclude the pebbles+boulders themselves. (Grain-size values in Table 3 are sub-millimeter). Do the clay and silt classes (pg. 5, lines 18-

10 19) have significant fractions of pebbles+boulders, which is only mentioned for sand (line 21)? If so, what are the ranges of that fraction for each class? Do pebbles+boulders significantly influence the bulk(?) physical properties in Table 3? Both clast and matrix play crucial roles in sediment deformation in Evans et al.'s comprehensive review (referenced here: Earth-Sci. Rev., 2006), especially their sections 5.1 to 5.4 on the "till-matrix framework".

We acknowledge that this section is confusing as written. We went back to some of the original papers discussing the composition of till to try and clarify the meaning of the descriptions. We have rewritten the first paragraph of section 2.3 to be as

The map of generalized grain size of glacial sediments is shown on Figure 2. A glacial sediment, diamiction or till (the later has a definitive glacial origin) is an unsorted material with grain size ranging from clay to boulder. Glacial sediments generally have a bimodal grain size distribution, with peaks in the course (pebble to boulder) and fine (clay to sand) fractions (Dreimanis and Vagners, 1971). The relative amount of course to fine is dependent on the distance from the source of the course material, so on glacial geology maps and datasets, glacial sediments are described in terms of the fine fraction only. To simplify the classification, we only have three main classification types, based on the dominant grain size of fine fraction. This classification scheme is based on the Surficial Materials in the Conterminous United States map (Soller and Reheis, 2004), and we attempted to unify this scheme with maps and data in Canada. The grain size of the sediments tends to have geographical dependence. As an example, in the map by Soller and Reheis (2004), clay rich glacial sediment exists in areas around the Great Lakes, where source material was derived from lake sediments, and sandy glacial in mountainous regions where there are extensive rock outcrops. The relative fraction of the sediment that is coarser than sand is not possible to quantify, since most of the data sources only give qualitative descriptions of the coarse fraction.

30 Uncertainty. In creating the dataset the authors necessarily and reasonably had to make some interpretations and interpolations between sources of data. However, there doesn't seem to be any record of where gaps have had to be filled or boreholes consulted. A map of data coverage (boreholes, geology map location etc), or another map showing some sort of confidence level in the data would be very useful for those using the data in model experiments and for focusing future work.

This is a really good suggestion. It took some time go back to the original shapefiles to create this, but we now include this in the final version of the dataset, and added this figure to the main document:

Figure 1. Data coverage (brown areas) derived directly from surficial geology maps. (a) sediment distribution (b) sediment grain size

5 Some notes on the data format that the data is provided in would be useful. As mentioned above, I could not check as not yet accessible through the repository.

We added these details to the last paragraph of section 2.1:

The final dataset is presented as shapefiles that are compatible with GIS programs, as well as 5 km resolution NetCDF files.

**10 4 Minor comments**

15

Abstract, page 1, line 5. Please state the scale or resolution of the dataset.

We added the scale here. (1:5 000 000 scale)

P1, L 17 - I find the use of the word substrate odd here, it is more about whether sediments were present beneath the ice sheet or not. To me, these sediments (or bedrock) would then be the substrate, i.e. the bit in contact with the bed of the ice is the substrate.

We reworded the sentence as follows:

Temperate ice sheets, such as the Laurentide and Eurasian ice sheets behaved differently depending on whether or not there was thick, continuous unconsolidated sediments underneath the ice (Clark and Walder, 1994).

P1, L19 - Consider adding the more recent reference of Storrar et al., 2014 on eskers across the Laurentide.

We have added the reference.

P2, L9 - Unclear subject matter. Presence or absence of what?

Added " of available unconsolidated sediment " to this sentence.

5 P2, L 15-17 - This section needs better linking to scope of the paper. These are all important factors, but some better crafting of the paragraph is required to state why we need to know about these things. In particular, here the subject jumps from the Laurentide to Svalbard without any linking.

This was added here to state that the conditions that probably existed on the Laurentide ice sheet is also applicable to modern glaciers. But perhaps such details are elaborated better in the subsequent paragraphs. We have removed these sentences.

10 P3, L 6-9 - These sentences are better incorporated into the following section.

We moved the paragraph to the next section.

*P3*, *L*13 - use of word "extended" is technically correct, but I wonder if better for reader if you use occurred between or similar, given the use of the word "extent" later to refer to where the ice got to.

We changed "extended" to a proper chronological descriptor word "happened".

15 P3, L 32 - requires rewording. Perhaps "information" rather than "glacial"

Thank you for pointing out the wording mistake. We changed it to read "glacial geological units".

Section 2.1. - A statement on the intended use and resolution of the data would be useful for those intending to use the dataset and to prevent misuse. I imagine the datasets will be useful for those doing ice sheet-scale experiments. However, the resolution may limit utility for those interested in a single outlet glacier/ice stream for example.

20 This is a good idea, we have added the following sentences:

We want to emphasize that these datasets are low resolution, generalized representations of geological properties. The intended use is for relatively low resolution ice sheet simulations (*i.e.* 5 km or great), and are not likely to be appropriate for resolving higher resolution features.

**P5, L 30 - No notes on clay**

25 We do not use a clay unit when inferring properties from geological maps. We have added the following sentence to emphasize this:

Since the distribution of clay rich till appears to correlate strongly with the location of lakes, it is not included.

P6, Section 2.5. This section would be useful for including the notes/map of "uncertainty" stated above.

As mentioned earlier, we now include a figure for data coverage, and included the shapefiles in the dataset.

P7, L 23. I think it worth restating here for the audience that your aim is not to draw specific conclusions about the form of the Laurentide in this paper. The following sections (3.2.1 to 3.2.3) do mention specifics of the modelled ice sheet. However, I think that these are safe as they fall short of evaluating whether there is an improvement or not, by just stating that there is a

5 *change induced by the data.*

We addressed this by revising section 3.1, as elaborated in the comments to Reviewer #1.

Additional references: Storrar, R.D., Stokes, C.R. and Evans, D.J., 2014. Morphometry and pattern of a large sample (> 20,000) of Canadian eskers and implications for subglacial drainage beneath ice sheets. Quaternary Science Reviews, 105, pp.1-25.

**References**

Clark, P. U. and Walder, J. S.: Subglacial drainage, eskers, and deforming beds beneath the Laurentide and Eurasian ice sheets, Geological Society of America Bulletin, 106, 304–314, https://doi.org/10.1130/0016-7606(1994)106<0304:SDEADB>2.3.CO;2, 1994.

Dreimanis, A. and Vagners, U. J.: Bimodal distribution of rock and mineral fragments in basal tills, in: Till, a symposium, edited by Goldth-

- 5 wait, R. P., pp. 237–250, Ohio State University Press, Columbus, Ohio, 1971.
- Soller, D. R. and Reheis, M. C.: Surficial Materials in the Conterminous United States, U.S. Geological Survey Open File Report OFR-03-275, U.S. Geological Survey, https://pubs.er.usgs.gov/publication/ofr2003275, scale 1:5,000,000, 2004.

---

## Author Response (AR2)

**Response to reviewers and revised manuscript**

Evan J. Gowan[1], Lu Niu[1], Gregor Knorr[1], and Gerrit Lohmann[1]

[1]Alfred Wegener Institute, Helmholtz Centre for Polar and Marine Research, Bremerhaven, Germany

**Correspondence:** Evan J. Gowan (evan.gowan@awi.de)

**Response to Topical Editor, Reinhard Drews**

We would like to thank Dr. Drews for editing our paper. Here is our response to the comments. The original comments are in italics, and our response is in normal font.

*Change last two sentences in abstract:*

5  *In order to demonstrate the importance of the basal boundary conditions for ice-sheet modelling, we changed the shear friction angle to account for a weaker substrate and found up to 40% changes in ice thickness compared to a reference run. Although incorporation of the ice-bed boundary remains model-dependent, our dataset provides an observational baseline for improving a critical weakness in current ice-sheet modelling.*

We have changed these two sentences to be as suggested.

10  *P4, l 16f not sure if the square meters have to be stated in addition to the square kilometers. I suggest removal.*

We have removed the bracketed conversion to meters.

*Figure 1 and 3: I would prefer a grey (rather than pink) bedrock.*

Pink is a pretty universal colour to represent crystaline bedrock on geology maps, so we would prefer to keep it as is.

*Figure 2: Scale bar and axis labels are missing.*

15  These have been added

*Revise 3.1 Change section heading to something more meaningful than "Parameters"*

We changed it to read "Geological parameters and impact on ice sheets"

*Conclusions: A large amount of work has gone into compiling this dataset. The conclusions don't convey that. In fact they read a little as if you ran out of breath.*

*Consider adding a few sentences in order to spice up the conclusions, and keep in mind that this section will be (after the abstract) the most widely read. Suggestions for inclusion are things like: "This dataset is a compilation out of XX geologic studies conducted between 19XX and 20XX and covers all of North America, Greenland and surrounding areas. We merged and optimized these different datasets for usage in low-resolution, long-timescale ice-sheet modeling. Using a specific model setup as an example, we show that significant thickness differences (up to 40 %) occur depending on which sediment type is prescribed at the ice-bed interface. This new dataset improves previously existing dataset by A,B and C and will be an important for better understanding processes at the ice-bed boundary. This is a critical parameter for the reconstruction of past ice sheets and hence for our understanding of past sea level variations (or whatever you think is the most important application)... "*

We have fleshed out the conclusions as follows:

Our compilation represents the first publicly available continuous sediment properties dataset that can be implemented into ice sheet modelling studies. We have presented three datasets that present different types of geological data, including sediment distribution, grain size, and bedrock geology for the regions in North America, Greenland and Iceland that were glaciated during the late Quaternary. The compilation directly incorporates information from over 50 maps and GIS datasets, plus additional information from over 40 other sources. These datasets are intended for use in ice sheet models, where the geological parameters will have impacts on ice sheet dynamics and hydrology. We demonstrated that changing the basal conditions in an ice sheet model on the basis of these datasets do impact the modelled thickness of the ice. In our simple experiments where we changed the shear friction angle to account for changes in geological properties based on inferred weakness of the ice-bed interface, there were changes of ice thickness by up to 40%. With these datasets, we hope that improvements can be made to ice sheet models to incorporate this geological data and create a more realistic representation of basal conditions. Examples of such application include changing the shear friction angle in a Mohr-Coulomb plastic basal sliding model, or to change water routing properties in a basal hydrology model. These properties are key to explain observed ice sheet dynamics, notably the rapid advance and retreat of the Laurentide Ice Sheet, during the last glacial cycle.

**Introduction**

We would like to thank Dr. Ely and an anonymous reviewer for reviewing the manuscript. Below is our responses to the reviewer comments and the steps we took to address their comments. Attached at the end is the manuscript, marked with the changes made.

5 One main thing is that we changed the title of the paper to "Geology datasets in North America, Greenland and surrounding areas for use with ice sheet models" to reflect the fact that Greenland, Iceland and surrounding areas are also included in the dataset.

We also revised the dataset to include a sediment distribution dataset for Saskatchewan by Simpson (1997). The figures have been changed to reflect this (relatively minor) change. We also have now included data coverage shapefiles to the dataset, to
10 show where there is mapped grain size and distribution information directly from surficial geology maps. Figures showing this are included in the main manuscript.

**Response to reviewer #1**

We would like to thank the reviewer for his or her or their comments. We have responded to the comments below. The comments from the reviewer is in italics, while our response is in normal font.

**General comments**

5    *This paper presents three geologic maps spanning northern North America and Greenland, showing quantities related to sediment at the ice-bed interface. As discussed, sediment has important effects on basal sliding and hydrology in ice sheet models of Quaternary glacial cycles. The effect of these maps is illustrated using the PISM ice sheet model, comparing simulations at LGM to those using uniform sediment properties. The geological data sources, methods of assimilation and resulting maps are well described, although that is mostly outside my experience, and the points below are from an ice-sheet modeling perspective.*

10    *The datasets themselves (Figs. 1-3) may be a worthwhile contribution for this area of ice-sheet modeling. However, the relationship to model parameters and usage presented in the paper is not as useful. An alternative would be not to include any linkage at all with physics and modeling, and just to present Figs. 1 to 3 as descriptions of the real world, and leave it to users to make the links to ice-sheet models. But the former is included here (Table 3, Eq. (1) and the PISM runs), and its physical basis could be improved and made more useful, as discussed below.*

15    In creating this paper, we had discussions about the appropriateness of including some ice sheet simulations using the datasets in this paper. The target of this journal is purely to describe the datasets, which could then be used by ice sheet modellers to improve their ice sheet simulations. The lead author (Evan J. Gowan) decided to include the simulations with the purpose of demonstrating "if you change the basal conditions on the basis of these maps, there will be an impact on resulting ice sheet, so they should be considered". We feel a more thorough analysis on the parameters that should be chosen is beyond the

20    scope of the paper. The default PISM model does not have the capabilities to implement the various parameters described in the paper (namely, discontinuous sediment cover), and were only included here as a guide as to what could be implemented into the ice sheet model if the maps presented here are to be used. The values of phi, with the except of the sediment grain size maps, were selected only to show contrasts between the different map units, on the basis of expected impact on the ice sheet, and do not have any physical meaning. The sediment distribution, for instance, should be parameterized in a way that

25    sediment deformation no longer has as much impact on the ice sheet dynamics. In PISM, there is no way to do this (although as mentioned late in the paper, we are in the process of incorporating this).

Upon reading the ice sheet modelling section, we realize that we were not explicit enough with the purpose of the ice sheet modelling (since both the reviewers have brought this up). Although we attempted to articulate this in third paragraph of section 3.2, we have rewritten the introductory paragraph in section 3.2 to clarify our intentions:

To show the utility of the dataset, we incorporate the information for use with the ice sheet model PISM 1.0 (Bueler and Brown, 2009; PISM authors, 2017), with the addition of an index forcing scheme described in Niu et al. (2017). In the standard version of PISM, the model for basal sliding has an assumption that there is a continuous layer of sediments underlying the ice sheet. Obviously, in areas where sediment coverage is discontinuous, this is not a valid model. Therefore the purpose of the following simulations are simply to demonstrate that if there is a contrast in the basal conditions based on the underlying geological parameters, there will be an impact on the resulting ice sheet simulation. The simulations are not necessarily reflective of actual basal conditions of the ice sheet.

**Specific comments**

*1. A Coulomb rheology model is adopted as used in the PISM model (Eq. 1), and one parameter (shear friction angle phi) is chosen to be related to the geologic quantities mapped here. There is little discussion (section 3.1, pg. 7) on the physics of how the map properties in Figs. 1-3 are related to the model's phi, and why phi is chosen over other model parameters such as those involved with N for instance (in Eq. 1, involving basal hydrology). The values chosen for the range of phi (10, 20, 30 degrees, pg. 8 lines 28-29) are not justified well; a reference could be given for how those relate to extremes of hard bed vs. deformable sediment (see (5) below). More discussion would be helpful on how the 3 quantities mapped here (sediment distribution, grain size, bedrock type) relate physically to sliding, why three maps are useful rather than one, and how they relate to other parameters in sliding models and not just phi. As a bonus, relations to Weertman sliding coefficients could be provided as well as to Coulomb parameters.*

As mentioned above, the Coulomb rheology model that we use in these simulations is not realistic to describe the actual basal conditions of the ice sheet. It does not truly describe a hard versus soft bed situation, since it does not include parameters on how sediment distribution affects ice sheet flow (as mentioned in paragraph 2 in section 3.1), or how bedrock could drain water underneath the ice sheet (as mentioned in paragraph 3 of section 3.1). As mentioned above, these simulations are only to illustrate that if you change the basal conditions on the basis of these maps, there is an impact on the ice sheet.

For the sediment distribution maps, we justified the parameters (see lines 28-30 on page 8) by setting "hard bedded" regions with the rock classification to have a high shear angle (which increases the strength of the base), and a low value where there is continuous sediments where it would be suspected to be weak.

The sediment grain size dataset used values that were based on measured values of the shear friction angle (lines 6-8 on page 9)

In the geology dataset, we changed the friction angle based on how easily the bedrock can erode to produce glacial sediments (lines 17-19 on page 9).

We do not wish to comment on Weertman parameters, as it is not clear that the Weertman sliding law is valid (Fowler, 2010), and there are few observations of what the bed roughness parameter is to attempt such a discussion (see section 7.2.2.1 in Cuffey and Paterson (2010)). The ice sheet model we use (PISM), also does not include a Weertman sliding law.

*2. In section 2.3 discussing sediment properties, the text says "Glacial sediments tend to be very poorly sorted, so these values should be assessed as being an average composition" (pg. 5 lines 12-13). Confusingly, the rest of the section seems to discuss grain-size properties only of the fine-grained material (matrix) that enclose pebbles and boulders (clast), and exclude the pebbles+boulders themselves. (Grain-size values in Table 3 are sub-millimeter). Do the clay and silt classes (pg. 5, lines 18-19) have significant fractions of pebbles+boulders, which is only mentioned for sand (line 21)? If so, what are the ranges of that fraction for each class? Do pebbles+boulders significantly influence the bulk(?) physical properties in Table 3? Both clast and matrix play crucial roles in sediment deformation in Evans et al.'s comprehensive review (referenced here: Earth-Sci. Rev., 2006), especially their sections 5.1 to 5.4 on the "till-matrix framework".*

In retrospect, we agree that it is confusing that the discussion only refers to the fine components of the glacial sediment. Glacial sediment has a bimodal grain size distribution, no glacial geology map used in this compilation describes the coarse fraction. We agree, having a quantitative assessment of the fraction of the till that is coarser than sand size would be a useful parameter to know, as having more coarse material will make the till stronger. Quantitative measurements simply do not exist in most areas. The qualitative classifications are also dependent on the author/organization, which is why we simplified the dataset to just three unit types. We have revised the first paragraph of section 3.2 to emphasize that that grain size is referring only to the fine component of the sediment.

The map of generalized grain size of glacial sediments is shown on Figure 5. A glacial sediment, diamiction or till (the later has a definitive glacial origin) is an unsorted material with grain size ranging from clay to boulder. Glacial sediments generally have a bimodal grain size distribution, with peaks in the course (pebble to boulder) and fine (clay to sand) fractions (Dreimanis and Vagners, 1971). The relative amount of course to fine is dependent on the distance from the source of the course material, so on glacial geology maps and datasets, glacial sediments are described in terms of the fine fraction only. To simplify the classification, we only have three main classification types, based on the dominant grain size of fine fraction. This classification scheme is based on the Surficial Materials in the Conterminous United States map (Soller and Reheis, 2004), and we attempted to unify this scheme with maps and data in Canada. The grain size of the sediments tends to have geographical dependence. As an example, in the map by Soller and Reheis (2004), clay rich glacial sediment exists in areas around the Great Lakes, where source material was derived from lake sediments, and sandy glacial in mountainous regions where there are extensive rock outcrops. The relative fraction of the sediment that is coarser than sand is not possible to quantify, since most of the data sources only give qualitative descriptions of the coarse fraction.

*3. Another perhaps naive physical question: if sedimentary till has been transported over long distances under the ice in previous glacial cycles, wouldn't the sediment properties be related to parent bedrock type(s) far upstream, and not to the local bedrock type as assumed here? This seems to be addressed by lines 26-28 on pg. 5 referring qualitatively to Fulton (1989), but isn't large-scale sediment transport a major feature of Quaternary glaciations?*

Although it seems intuitive that glacial transport will strongly influence sediment composition, it actually isn't as large as one might think. For example, some of the lobes in the southern Laurentide region were sampled for zircons by Kassab et al. (2017), and they found that even though there was ice streaming from the Canadian Shield, the age of the zircons indicated that the source of the till still largely reflected the underlying Michigan Basin rocks. At any rate, the target of this paper is to represent the sediments for the last glaciation, in previous glaciations it was likely different.

*4. (related to (1) above). The spatial patterns of the 3 datasets in Figs. 1-3 are all quite similar. That is, roughly speaking they all show mainly one value for Hudson Bay and the outer regions of the former LGM Laurentide, and mainly another value for Canadian Shield regions surrounding Hudson Bay, the Canadian Rockies, and Greenland. So not surprisingly the patterns of model LGM delta(ice-thickness) sensitivities shown in Figs. 4d-6d are all quite similar to each other.*

*Although the patterns are similar, the magnitudes of phi in Figs. 4b-6b differ. For instance, the values over Hudson Bay vs. Canadian Shield regions are generally ~10 vs. 20 degrees in Fig. 4b using map #1, ~20 vs. 30 degrees in Fig. 5b using map #2, and ~10 vs. 30 degrees in Fig. 6b using map #3. It is left for ice-sheet modelers to choose which map is best for phi, for which no guidance is given.*

For the glacial sediment composition, the values of phi are probably well justified as it is based on actual values from (Cuffey and Paterson, 2010) (as noted in the text). For the rest of the properties, a completely different basal condition model should be used, as changes in basal conditions due to sediment distribution (for instance) are not based the till friction angle, but rather changes in the basal hydrological system. We are working on this (as noted on lines 14 and 15 on page 8 of the original manuscript). The experiments shown here are simply to show in a qualitative sense that changing the basal conditions on the basis of the maps affects the evolution of the ice sheet (as addressed above).

*5. Model ice-thickness results in Figs. 4-6 are compared with those using a uniform value of phi=30. The delta(thickness) magnitudes in the (d) panels seem smaller and much less extensive than in early Laurentide studies exploring sediment vs. hard-bed effects, such as Fisher et al. (1985) and Licciardi et al. (1998) referenced here on pg. 2, line 14. Perhaps this is due to smaller contrasts in yield stress here than in the earlier studies, and a larger range of phi could be used than 10 to 30 degrees.*

As noted on lines 9-13 on page 8 of the original manuscript, the impact on the ice sheet is limited only to areas where there is sufficient ice flow to cause melting at the base, and where that is not true, changing the shear friction angle has no effect. The figure added below for point #8 shows this. There is little change over most of the ice sheet because there is not enough water produced in the model to saturate the sediments. Further changing the shear friction angle in those regions will have no effect on the outcome. The range of shear friction angles used here fall within the range of realistic values for till, no geological material will have shear friction angles less than $10°$. This is why we are working on creating a new basal conditions model to better incorporate the geological parameters in a realistic way (as noted on lines 14 and 15 on page 8).

*6. In several places the text mentions or implies stratigraphic relationships between the underlying bedrock, layers of glacially derived sediment, perhaps layers of earlier (or Holocene?) non-glacial sediment, and water bodies; for instance on pg. 3 lines 1-3, pg. 4 lines 1-4, pg. 4 line 19. For a non-geologist, it is challenging to conceptualize this clearly, and to keep straight what*

*the stratigraphic relationships are (for glacial vs. non-glacial sediment), and whether they are important. I suggest slightly expanded discussion in places. Also a simple diagram would be helpful that shows the various idealized stratigraphic cases, with labeled horizontal rectangles one on top of each other. The cases would always have underlying bedrock, then one case with bedrock covered by glacial sediment (perhaps with horizontal breaks indicating patchiness), another with bedrock +*

5  *non-glacial sediment, another with bedrock + glacial sediment + water body, etc.*

We have added the following illustration showing the relationship between bedrock, glacial sediments, water bodies, and post-glacial sediments:

[Figure]

**Figure 1.** Illustration showing the relationship between the bedrock, glacial sediments and postglacial sediments. In glacial times, the ice sheet is in contact with glacial sediments created by the ice sheet itself, and bedrock. In post-glacial times, the bedrock and glacial sediments can be obscured by water bodies and post-glacial sediments.

We have added the following figure which shows the relationship between the sediment and bedrock in the distribution classes:

[Figure]

**Figure 2.** Illustration of how the sediment distribution relates to the underlying bedrock and thickness of the sediments. The rock class has only isolated patches of sediment, the veneer class has a thin sediment layer with bedrock outcrops and a visible influence of bedrock topography on the surface, while with the blanket class, the sediments completely obscure the bedrock surface.

*7. If possible, ranges of thickness values for each type of layer could be shown in the diagram suggested above. These are mentioned in places in the text, and could be gathered there. The paper says (pg. 7 line 14) "it is not possible to give a detailed quantitative estimate of (sediment) distribution and thickness", but even rough ranges would be helpful. Melanson et al. (QSR, 2013, Fig. 1), and Hildes et al. (QSR, 2004, Fig. 4) produced sediment thickness maps based on Fulton et al. (2005) referenced here; could a guide be provided here? Maps of sediment thickness are not so important for models of LGM or the last glacial cycle, but are more helpful for longer-term studies of Plio-Pleistocene variations involving sediment evolution, such as Ganopolski et al., Clim. Past, 2011.*

For the distribution dataset, the important parameter is not necessarily sediment thickness (although we have included some quantitative values in the description of the units), but rather the percentage of surface is covered in sediments versus bare rock. It would be difficult to put hard numbers on this, since many surficial geology maps only give qualitative categories, and the definition also varies depending on the author of the map (as mentioned in the text of section 2.2). To emphasize this point, we have added the following to the intro paragraph of section 2.2:

Many maps used in this dataset only give qualitative descriptions of the distribution, and the definition often varies between mappers. As a result, it is not possible to give an exact range for sediment thickness or percentage sediment cover. We recommend modellers explore a range of values.

The classification scheme used by Melanson et al. (2013) and Hildes et al. (2004) for defining the distribution of sediments is essentially the same as ours, which is to say it is a qualitative assessment of sediment distribution. We have added the following to section 2.2:

A scheme similar to this has been used in the studies by Hildes et al. (2004) and Melanson et al. (2013) for use in the modelling of sediment transport. The difference in our dataset is that we explicitly do not include post-glacial sediments, and instead try to fill these gaps with supplemental information.

As for thickness maps, they produced these using an empirical formula based on those same qualitative descriptors. In order to get a quantitative assessment of sediment cover in longer-term studies, it would be necessary to first to have data on the amount of sediment has actually been transported glacially. From that, and inverse model could perhaps be applied. Such a dataset does not currently exist, but from some discussions the lead author (Evan Gowan) has had with geologists, such a dataset is under construction for some areas in the United States.

*8. The areas in Figs. 4d-6d where model LGM delta(ice thicknesses) are appreciable occur mostly around the margins of the ice sheet. The paper suggests this is controlled in part by areas where the bed is sufficiently warm and wet to allow sliding in the PISM model (pg. 8 line 35). This could be corroborated by showing a model map of basal temperature or basal melt rate*

5   *for 21 ka.*

We added the following plot that shows the basal heating.

[Figure]

**Figure 3.** Areas in the default simulation where basal frictional heating exceeds 0.01 W/m$^2$ (shown in red). The grey region is where there is grounded ice.

**Technical points:**

*pg. 5 lines 30-33: The classification relating bedrock to grain size is given for silt (line 30) and sand (line 32), but seems to be missing for clay.*

10   We did not include a clay unit here, because as mentioned on lines 14-16 on page 5, clay rich glacial sediments are likely derived from lake sediment, rather than bedrock.

*pg. 6 line 11: The name "Felsic volcanic" is used twice. Possibly the 2nd should be "Mafic volcanic".*

This has been fixed.

*pg. 7 line 4-6: Table 3 is discussed in section 3.1, before Eq. (1) and shear friction angle and cohesion (quantities in the table) are discussed in section 3.2. This could be resolved easily by referring to Eq. (1) and following text in the table caption.*

We added a reference to equation 1 to the table 3 title.

*pg. 7 lines 9-11. The two consecutive sentences beginning "The patchiness of sediment..." and "The lack of sediment.." seem to say nearly the same thing. Is the distinction one of horizontal scale?*

These two sentences refer to two different things. The first means that bedrock sticking out will resist flow due to increased friction, as the rock sticks into the glacier. The second sentence refers to the fact that the lack of sediments means there is no sediment deformation as a mechanism to flow. We modified the second sentence to make that clear.

> The lack of sediment in an otherwise sediment covered region may increase resistance to flow as well if sediment deformation is a dominant factor in controlling flow

*pg. 7 line 17: Regarding drainage of water under the ice, perhaps clarify in a few words that the permeability of bedrock under the sediment "aquifer" is involved here, I think. (As opposed to permeability of sediment as in Table 3, which would use map #2 for grain size and not map #3 for bedrock type).*

We appended "into the bedrock aquifer" at the end of the sentence.

*pg. 8 line 34; pg. 9 line 13; pg. 9 line 22: These sentences say that south of the Great Lakes at LGM, the ice sheet extends further than in the reference simulation. However there is no discernible difference in the margin locations south of the Great Lakes in Figs. 4a vs. c, 5a vs. c, 6a vs. c. How many km does "further south" mean in the sentences, and why is it not visible in the figures?*

We admit that you have to look very closely to identify that it indeed went further south. It is only one grid cell point, which is 20 km, which is visible by the blue area on the difference map. We have noted this in the text.

**Minor corrections/suggestions**

*pg. 1 line 4: Should be "distribution of surficial".*

Fixed.

*pg. 3 line 22: "When" should be "In".*

Using the word "when" is intentional.

*pg. 4 line 11: Is "bend simplify" intended?*

That is the name of the tool we used in ArcGIS.

*pg. 4 line 16; pg 5 line 9; pg. 7 line 4; pg. 9 line 9,15: "on" should be "in".*

Unless Copernicus' style guide requires this convention (and it doesn't appear to), we are leaving this as is.

*pg. 6 line 18: Should be "as most of these".*

Fixed

*pg. 7 line 11: Should be "of the latter". pg. 7 line 17: "later" should be "latter".*

Fixed.

**Response to Jeremy Ely (reviewer #2)**

We would like to thank Dr. Ely for his comments. We have responded to the comments below. The comments from the reviewer is in italics, while our response is in normal font.

**General comments**

*General comments: This paper presents useful datasets regarding the geology and sedimentology of North America, Greenland, Iceland and parts of Russia. The datasets are extensive and useful for the community. The demonstration of use in ice sheet models is interesting and also useful for the reader to consider how future experiments with this and similar datasets might be conducted. However, the aim of these simulations for this paper should be noted in the text (i.e. to demonstrate utility of the dataset, not to draw scientific conclusions about the Laurentide which I presume is the focus of a later paper). The authors should be applauded for citing all the original literature that goes into this dataset in Tables 1 and 2. It may also be useful to distribute a similar bibliography with the dataset. This is therefore a very worthwhile contribution to the literature and I hope that my comments below can help improve the manuscript.*

As mentioned in the response to reviewer #1, the intention of our simulations was just as mentioned here - to demonstrate the utility of the datasets. We have updated the section to make this more clear (see response to reviewer #1 for details). Adding the bibliography to the datasets is a good idea, we have added it to the dataset.

**Specific comments**

*The title - only North America is mentioned, yet the data includes Greenland and Iceland. I think these places need incorporating in the title somehow. As far as I can tell from figures (could not check the actual data as pangea is password protected) the whole of North America is not covered by the data either.*

We apologize for the inaccessibility of the datasets - Pangaea added a password for some reason. Once this was pointed out, we had the password removed.

This compilation started off as being purely focused on North America, the additions of Greenland and Iceland to the dataset came very late in the process. We will change the title to the following:

Geology datasets in North America, Greenland and surrounding areas for use with ice sheet models

*2. In section 2.3 discussing sediment properties, the text says "Glacial sediments tend to be very poorly sorted, so these values should be assessed as being an average composition" (pg. 5 lines 12-13). Confusingly, the rest of the section seems to discuss grain-size properties only of the fine-grained material (matrix) that enclose pebbles and boulders (clast), and exclude the*

*pebbles+boulders themselves. (Grain-size values in Table 3 are sub-millimeter). Do the clay and silt classes (pg. 5, lines 18-19) have significant fractions of pebbles+boulders, which is only mentioned for sand (line 21)? If so, what are the ranges of that fraction for each class? Do pebbles+boulders significantly influence the bulk(?) physical properties in Table 3? Both clast*
5 *and matrix play crucial roles in sediment deformation in Evans et al.'s comprehensive review (referenced here: Earth-Sci. Rev., 2006), especially their sections 5.1 to 5.4 on the "till-matrix framework".*

We acknowledge that this section is confusing as written. We went back to some of the original papers discussing the composition of till to try and clarify the meaning of the descriptions. We have rewritten the first paragraph of section 2.3 to be as follows.

10     The map of generalized grain size of glacial sediments is shown on Figure 2. A glacial sediment, diamiction or till (the later has a definitive glacial origin) is an unsorted material with grain size ranging from clay to boulder. Glacial sediments generally have a bimodal grain size distribution, with peaks in the course (pebble to boulder) and fine (clay to sand) fractions (Dreimanis and Vagners, 1971). The relative amount of course to fine is dependent on the distance from the source of the course material, so on glacial geology maps and datasets, glacial sediments
15     are described in terms of the fine fraction only. To simplify the classification, we only have three main classification types, based on the dominant grain size of fine fraction. This classification scheme is based on the Surficial Materials in the Conterminous United States map (Soller and Reheis, 2004), and we attempted to unify this scheme with maps and data in Canada. The grain size of the sediments tends to have geographical dependence. As an example, in the map by Soller and Reheis (2004), clay rich glacial sediment exists in areas around the Great Lakes,
20     where source material was derived from lake sediments, and sandy glacial in mountainous regions where there are extensive rock outcrops. The relative fraction of the sediment that is coarser than sand is not possible to quantify, since most of the data sources only give qualitative descriptions of the coarse fraction.

*Uncertainty. In creating the dataset the authors necessarily and reasonably had to make some interpretations and interpolations between sources of data. However, there doesn't seem to be any record of where gaps have had to be filled or boreholes*
25 *consulted. A map of data coverage (boreholes, geology map location etc), or another map showing some sort of confidence level in the data would be very useful for those using the data in model experiments and for focusing future work.*

This is a really good suggestion. It took some time go back to the original shapefiles to create this, but we now include this in the final version of the dataset, and added this figure to the main document:

*Some notes on the data format that the data is provided in would be useful. As mentioned above, I could not check as not yet*
30 *accessible through the repository.*

We added these details to the last paragraph of section 2.1:

     The final dataset is presented as shapefiles that are compatible with GIS programs, as well as 5 km resolution NetCDF files.

[Figure]

**Figure 4.** Data coverage (brown areas) derived directly from surficial geology maps. (a) sediment distribution (b) sediment grain size

**Minor comments**

*Abstract, page 1, line 5. Please state the scale or resolution of the dataset.*

We added the scale here. (1:5 000 000 scale)

*P1, L 17 - I find the use of the word substrate odd here, it is more about whether sediments were present beneath the ice sheet or not. To me, these sediments (or bedrock) would then be the substrate, i.e. the bit in contact with the bed of the ice is the substrate.*

We reworded the sentence as follows:

Temperate ice sheets, such as the Laurentide and Eurasian ice sheets behaved differently depending on whether or not there was thick, continuous unconsolidated sediments underneath the ice (Clark and Walder, 1994).

*P1, L 19 - Consider adding the more recent reference of Storrar et al., 2014 on eskers across the Laurentide.*

We have added the reference.

*P2, L 9 - Unclear subject matter. Presence or absence of what?*

Added " of available unconsolidated sediment " to this sentence.

*P2, L 15-17 - This section needs better linking to scope of the paper. These are all important factors, but some better crafting of the paragraph is required to state why we need to know about these things. In particular, here the subject jumps from the Laurentide to Svalbard without any linking.*

This was added here to state that the conditions that probably existed on the Laurentide ice sheet is also applicable to modern glaciers. But perhaps such details are elaborated better in the subsequent paragraphs. We have removed these sentences.

*P3, L 6-9 - These sentences are better incorporated into the following section.*

We moved the paragraph to the next section.

*P3, L 13 - use of word "extended" is technically correct, but I wonder if better for reader if you use occurred between or similar, given the use of the word "extent" later to refer to where the ice got to.*

We changed "extended" to a proper chronological descriptor word "happened".

*P3, L 32 - requires rewording. Perhaps "information" rather than "glacial"*

Thank you for pointing out the wording mistake. We changed it to read "glacial geological units".

*Section 2.1. - A statement on the intended use and resolution of the data would be useful for those intending to use the dataset and to prevent misuse. I imagine the datasets will be useful for those doing ice sheet-scale experiments. However, the resolution may limit utility for those interested in a single outlet glacier/ice stream for example.*

This is a good idea, we have added the following sentences:

> We want to emphasize that these datasets are low resolution, generalized representations of geological properties.
> The intended use is for relatively low resolution ice sheet simulations (*i.e.* 5 km or great), and are not likely to be appropriate for resolving higher resolution features.

*P5, L 30 - No notes on clay*

We do not use a clay unit when inferring properties from geological maps. We have added the following sentence to emphasize this:

> Since the distribution of clay rich till appears to correlate strongly with the location of lakes, it is not included.

*P6, Section 2.5. This section would be useful for including the notes/map of "uncertainty" stated above.*

As mentioned earlier, we now include a figure for data coverage, and included the shapefiles in the dataset.

*P7, L 23. I think it worth restating here for the audience that your aim is not to draw specific conclusions about the form of the Laurentide in this paper. The following sections (3.2.1 to 3.2.3) do mention specifics of the modelled ice sheet. However, I think that these are safe as they fall short of evaluating whether there is an improvement or not, by just stating that there is a change induced by the data.*

We addressed this by revising section 3.1, as elaborated in the comments to Reviewer #1.

*Additional references: Storrar, R.D., Stokes, C.R. and Evans, D.J., 2014. Morphometry and pattern of a large sample (> 20,000) of Canadian eskers and implications for subglacial drainage beneath ice sheets. Quaternary Science Reviews, 105, pp.1-25.*

**References**

Bueler, E. and Brown, J.: Shallow shelf approximation as a "sliding law" in a thermodynamically-coupled ice sheet model, J. Geophys. Res., 114, https://doi.org/10.1029/2008JF001179, 2009.

5   Clark, P. U. and Walder, J. S.: Subglacial drainage, eskers, and deforming beds beneath the Laurentide and Eurasian ice sheets, Geological Society of America Bulletin, 106, 304–314, https://doi.org/10.1130/0016-7606(1994)106<0304:SDEADB>2.3.CO;2, 1994.

Cuffey, K. M. and Paterson, W. S. B.: The physics of glaciers, Elsevier, 2010.

Dreimanis, A. and Vagners, U. J.: Bimodal distribution of rock and mineral fragments in basal tills, in: Till, a symposium, edited by Goldthwait, R. P., pp. 237–250, Ohio State University Press, Columbus, Ohio, 1971.

10   Fowler, A.: Weertman, Lliboutry and the development of sliding theory, Journal of Glaciology, 56, 965–972, https://doi.org/10.3189/002214311796406112, 2010.

Hildes, D. H., Clarke, G. K., Flowers, G. E., and Marshall, S. J.: Subglacial erosion and englacial sediment transport modelled for North American ice sheets, Quaternary Science Reviews, 23, 409–430, https://doi.org/10.1016/j.quascirev.2003.06.005, 2004.

Kassab, C. M., Brickles, S. L., Licht, K. J., and Monaghan, G. W.: Exploring the use of zircon geochronology as an indicator of Laurentide

15   Ice Sheet till provenance, Indiana, USA, Quaternary Research, 88, 525–536, https://doi.org/10.1017/qua.2017.71, 2017.

Melanson, A., Bell, T., and Tarasov, L.: Numerical modelling of subglacial erosion and sediment transport and its application to the North American ice sheets over the Last Glacial cycle, Quaternary Science Reviews, 68, 154–174, https://doi.org/10.1016/j.quascirev.2013.02.017, 2013.

Niu, L., Lohmann, G., Hinck, S., and Gowan, E. J.: Sensitivity of atmospheric forcing on Northern Hemisphere ice sheets during the last

20   glacial-interglacial cycle using outputs from PMIP3, Climate of the Past Discussion, https://doi.org/10.5194/cp-2017-105, in review, 2017.

PISM authors: PISM, a Parallel Ice Sheet Model, http://www.pism-docs.org, 2017.

Simpson, M. A.: Surficial Geology Map of Saskatchewan, Map, Environment Branch, Saskatchewan Research Council, from surficial geology maps at 1:250 000 scale by J.E. Campbell and M.A. Simpson and 1:1 000 000 scale by B.T Schreiner, scale: 1:1,000,00, 1997.

Soller, D. R. and Reheis, M. C.: Surficial Materials in the Conterminous United States, U.S. Geological Survey Open File Report OFR-03-275, U.S. Geological Survey, https://pubs.er.usgs.gov/publication/ofr2003275, scale 1:5,000,000, 2004.